# CXCR4 engagement triggers CD47 internalization and antitumor immunization in a mouse model of mesothelioma

Rosanna Mezzapelle[1,2], Francesco De Marchis[1] (ID), Chiara Passera[1], Manuela Leo[3], Francesca Brambilla[1], Federica Colombo[1,4], Maura Casalgrandi[5], Alessandro Preti[5], Samuel Zambrano[1,2], Patrizia Castellani[6] (ID), Riccardo Ertassi[1], Marco Silingardi[2], Francesca Caprioglio[2], Veronica Basso[7], Renzo Boldorini[8,9], Angelo Carretta[10], Francesca Sanvito[11], Ottavio Rena[12], Anna Rubartelli[6], Lina Sabatino[3], Anna Mondino[7], Massimo P Crippa[1], Vittorio Colantuoni[3,*] & Marco E Bianchi[1,2,*]

## Abstract

Boosting antitumor immunity has emerged as a powerful strategy in cancer treatment. While releasing T-cell brakes has received most attention, tumor recognition by T cells is a pre-requisite. Radiotherapy and certain cytotoxic drugs induce the release of damage-associated molecular patterns, which promote tumor antigen cross-presentation and T-cell priming. Antibodies against the "do not eat me" signal CD47 cause macrophage phagocytosis of live tumor cells and drive the emergence of antitumor T cells. Here we show that CXCR4 activation, so far associated only with tumor progression and metastasis, also flags tumor cells to immune recognition. Both CXCL12, the natural CXCR4 ligand, and BoxA, a fragment of HMGB1, promote the release of DAMPs and the internalization of CD47, leading to protective antitumor immunity. We designate as Immunogenic Surrender the process by which CXCR4 turns in tumor cells to macrophages, thereby subjecting a rapidly growing tissue to immunological scrutiny. Importantly, while CXCL12 promotes tumor cell proliferation, BoxA reduces it, and might be exploited for the treatment of malignant mesothelioma and a variety of other tumors.

**Keywords** CD47; CXCR4; HMGB1; immunogenic cell death; mesothelioma
**Subject Categories** Cancer; Immunology

## Introduction

Chronic inflammation and the presence of an unfavorable inflammatory microenvironment can promote tumor development. A common example is colon carcinoma (Terzić *et al*, 2010), but as representative is malignant mesothelioma (MM), a tumor that is associated with asbestos exposure and comprises a large inflammatory component, in particular macrophages (Lievense *et al*, 2013). We previously showed that both MM cells and macrophages secrete High Mobility Group Box 1 protein (HMGB1) (Yang *et al*, 2010; Jube *et al*, 2012), an alarmin that alerts the innate and adaptive immune systems to tissue damage and cell stress (Bianchi *et al*, 2017). HMGB1 plays a central role in tissue regeneration, in part by recruiting monocytes/macrophages via the CXCR4 receptor and directing them toward a tissue-healing phenotype (Tirone *et al*, 2018). In MM, secreted HMGB1 sustains chronic inflammation initially caused by asbestos and supports disease progression (Jube *et al*, 2012; Xue *et al*, 2020). HMGB1 has several receptors, among which TLR4, RAGE, and CXCR4 are the most well-known (Bianchi *et al*, 2017). BoxA is a fragment of HMGB1 that corresponds to its first HMG-box

---

1   Chromatin Dynamics Unit, Division of Genetics and Cell Biology, IRCCS San Raffaele Scientific Institute, Milan, Italy
2   School of Medicine, Vita-Salute San Raffaele University, Milan, Italy
3   Department of Sciences and Technologies, University of Sannio, Benevento, Italy
4   Department of Electronics, Information and Bioengineering, Politecnico di Milano, Milano, Italy
5   HMGBiotech S.r.l., Milan, Italy
6   Cell Biology Unit, IRCCS Ospedale Policlinico San Martino, Genova, Italy
7   Division of Immunology, Transplantation and Infectious Diseases, Lymphocyte Activation Unit, IRCCS San Raffaele Scientific Institute, Milan, Italy
8   Department of Health Science, School of Medicine, University of Eastern Piedmont Amedeo Avogadro, Vercelli, Italy
9   Pathology Unit, Maggiore della Carità Hospital, Novara, Italy
10  Department of Thoracic Surgery, IRCCS San Raffaele Scientific Institute, Milan, Italy
11  Department of Pathology, IRCCS San Raffaele Scientific Institute, Milan, Italy
12  Unit of Thoracic Surgery, Maggiore della Carità Hospital, Novara, Italy
    *Corresponding author. Tel: +39 0824 305102; E-mail: vittorio.colantuoni@gmail.com
    **Corresponding author. Tel: +39 0226 434762; E-mail: bianchi.marco@hsr.it

domain and competes with HMGB1 for binding to the RAGE and TLR4 without activating them (Venereau *et al*, 2016; He *et al*, 2018). We previously reported that targeting HMGB1 with monoclonal antibodies or BoxA extends the survival of mice xenografted with human MM cells by interfering with tumor cell proliferation (Yang *et al*, 2015). However, extracellular HMGB1 also primes antigen recognition (Rovere-Querini *et al*, 2004) and is involved in immunogenic cell death (ICD). ICD is induced by certain chemotherapeutics or radiotherapy and increases the processing of apoptotic tumor cells by dendritic cells (DCs), enhances their immunogenicity, and elicits an efficient antitumor immune response and immunological memory (Galluzzi *et al*, 2020) The mechanism of ICD involves the apoptosis of tumor cells, preceded by endoplasmic reticulum (ER) stress, with concomitant induction of the unfolded protein response (UPR) and the release of HMGB1, ATP, and calreticulin (Kroemer *et al*, 2012). Calreticulin is an abundant ER-resident protein that becomes an "eat me" signal once relocated to the cell surface (ecto-calreticulin).

To test whether targeting HMGB1 is beneficial or detrimental in immunocompetent tumor-bearing hosts, we set up a syngeneic model of MM, where mouse AB1 malignant mesothelioma cells are grafted into the peritoneum of syngeneic BALB/c mice (Mezzapelle *et al*, 2016). Surprisingly, we found that BoxA, besides being antiproliferative, also promotes protective antitumor immunity responsible for MM rejection and long-term survival in a large fraction of mice.

Exploration of the mode of action of BoxA revealed that it acts via CXCR4. CXCR4 is a G-protein coupled receptor that induces cell migration upon binding its main ligand CXCL12 (also known as SDF-1) (Teicher & Fricker, 2010; Bianchi & Mezzapelle, 2020). CXCR4 is also involved in metastatisation, and in many types of tumors upregulation of CXCR4 and of its ligand CXCL12 are predictive of short disease-free survival (Guo *et al*, 2016). Robust upregulation of CXCR4 was reported in human mesothelioma cell lines and in mesothelioma tissues (Li *et al*, 2011). However, in MM cells engagement of CXCR4 by BoxA does not promote cell growth but rather induces the surface exposure of calreticulin and the depletion of surface CD47, tilting the balance of "eat me" and "don't eat me" signals, and promoting tumor cell phagocytosis by macrophages. CD47 is a ubiquitous transmembrane protein that prevents the phagocytosis of functionally fit cells by interacting with its ligand SIRP1α (signal regulatory protein 1α) on the surface of macrophages and DCs (Barclay & van den Berg, 2014). Lack of CD47 on erythrocytes, platelets, and lymphohematopoietic cells results in rapid clearance of these cells by macrophages (Blazar *et al*, 2001). CD47 is expressed at increased level on the cell surface by a variety of malignant cells (Willingham *et al*, 2012); its blockade with monoclonal antibodies allows the efficient phagocytosis of cancer cells and leads to tumor rejection and development of antitumor immunity (Liu *et al*, 2015). CD47 blockade has remarkable therapeutic efficacy in various preclinical models of bladder, colon and breast cancer, glioblastoma, lymphoma, and acute lymphocytic leukemia (Jaiswal *et al*, 2009; Willingham *et al*, 2012; Liu *et al*, 2015). The published studies involve the masking of CD47 by antibodies that prevent its interaction with SIRP1α; whether and how CD47 exposure is modulated in response to the microenvironment is still unknown. Here, we show that CXCR4 engagement promotes the internalization of CD47 and the downstream antitumor responses, both when triggered by BoxA or CXCL12. Thus, we argue that the

CXCL12/CXCR4 axis activates immunosurveillance via a mechanism (which we name Immunogenic Surrender) that allows tumor identification by innate cells and tumor-specific T-cell priming.

# Results

## BoxA promotes tumor rejection and the development of protective antitumor immune memory

HMGB1 promotes human MM cell survival and proliferation via RAGE (Jube *et al*, 2012), whereas BoxA, its N-terminal fragment, acts as an HMGB1 competitor and antagonist on RAGE and TLR4 receptors (Venereau *et al*, 2016). Accordingly, BoxA was found to reduce tumor growth and extend mice survival in a model where human MM cells were injected into immunodeficient mice (Jube *et al*, 2012; Yang *et al*, 2015). However, HMGB1 plays a key role in inducing ICD (Kroemer *et al*, 2012), and therefore, targeting HMGB1 might reduce antitumor immune responses. To investigate the antitumor potential of BoxA in immune-competent mice and possible underlying mechanisms, we exploited the syngeneic mouse model of MM we previously developed, where mouse AB1-B/c MM cells are engrafted in the peritoneum of BALB/c mice (Mezzapelle *et al*, 2016). Inoculation of $7 \times 10^4$ MM cells produced MM tumors (Fig 1A) that were highly infiltrated by inflammatory cells, mostly represented by CD206[+] CD86[−] macrophages, and few CD3[+] T and B cells (Fig EV1A and B). HMGB1 was highly expressed both in the nucleus and in the cytosol of tumor cells. This pattern is very similar to that of human mesothelioma (Fig EV1C).

In a first small-scale experiment, we inoculated AB1-B/c mouse MM cells in the peritoneum of 12 BALB/c mice, and 3 days later, we started treatment with 0, 200, 400, and 800 μg BoxA, three times a week. After 22 days, all control mice had developed tumor lesions, while mice treated with 800 μg BoxA had no discernible lesions, and mice treated with smaller doses of BoxA had an intermediate incidence (Fig 1B). Thus, BoxA was not toxic at the highest dose (800 μg per injection) and showed antitumor effects also in immunocompetent mice.

To follow tumor development in longitudinal analysis by BioLuminescence Imaging (BLI), we adopted AB1 cells expressing luciferase (AB1-B/c-LUC, henceforth called MM cells). Mice received i.p. delivery of BoxA or PBS (control) 2 days after MM injection (Fig 1A). At day 6, the high level of abdominal BLI signal (Fig 1C) indicated the engraftment of MM cells in both control and BoxA-treated mice. In the following weeks, most control mice experienced an increase of the BLI signal and had to be sacrificed. Notably, 4 control mice (20%) showed a tenfold decrease of the BLI signal relative to the first measurement, but then experienced remission and survived beyond the end of the 3-week treatment (Fig 1C and D). In contrast, 18 BoxA-treated mice showed a decrease of the BLI signal after day 6, in some cases to 10-fold below the initial measurement, and 15 (75%) survived after the end of the treatment. At day 75, the difference in survival curves between control mice and BoxA-treated mice was highly significant ($P < 0.0001$) (Fig 1D). We sacrificed two of the surviving mice per group (control and BoxA treated) and we could not identify any tumor mass, either in the abdomen or elsewhere; this difference in tumor rejection was highly significant (4/20 versus 15/20, $P = 0.0012$ Fisher's test).

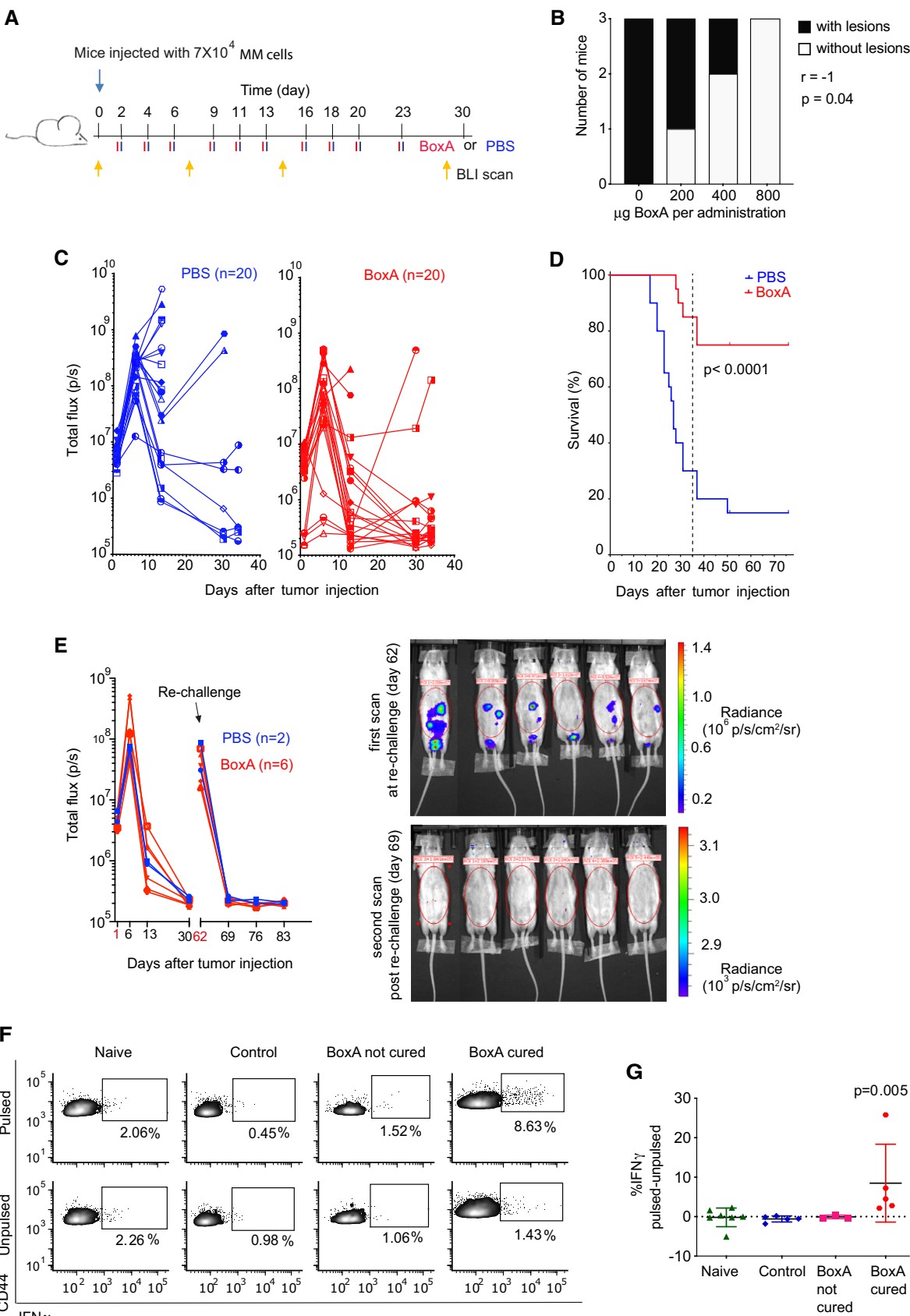

**Figure 1.**

**Figure 1.  BoxA increases survival and induces immunization in a syngeneic mouse model of mesothelioma.**

A  Scheme of the experiment. BALB/c mice were inoculated i.p. with $7 \times 10^4$ MM cells and treated with either 800 μg BoxA (red bars) or PBS (blue bars) three times a week, 10 times in total. Yellow arrows represent BLI imaging.

B  Treatment with BoxA at increasing doses reduced the number of mice with detectable tumor lesions in a statistically significant dose-dependent manner (Spearman correlation: $P = 0.04$). In this experiment, no BLI measurement was taken.

C  Forty mice were inoculated with MM cells and treated or not with BoxA. Tumor growth was detected via BLI. Lines that do not reach day 34 correspond to mice that were sacrificed for ethical reasons.

D  Kaplan–Meier survival curves. Statistics: log-rank Gehan–Breslow–Wilcoxon test, $P < 0.0001$, $n = 20$ per group.

E  Tumor growth after MM re-challenge detected by BLI.

F–G  Flow cytometry analysis of luciferase-specific T cells. In (F), representative dot plots depict intracellular IFNγ levels in gated $CD8^+$ $CD44^{high}$ T cells. In (G), differences in the percentage of IFNγ producing $CD8^+$ $CD44^{high}$ cells, pulsed relative to unpulsed. Statistics: Kruskal–Wallis test; each dot represents a mouse ($n = 3$–7 per group); bars represent mean ± SD.

Source data are available online for this figure.

These results indicate that BoxA treatment can extend the survival of model MM mice, but most of all increases the fraction of mice that reject the tumor. Notably, the efficacy of BoxA appears strikingly higher in immunocompetent mice compared to immunodeficient ones (Yang *et al*, 2015), suggesting that the immune system is an active player in the activity of BoxA.

To test whether surviving mice had developed immunological memory against the tumor, we re-challenged them with MM cells. All mice showed a high level of bioluminescence soon after the re-challenge, but only background levels 7 days later; all of them survived for several weeks without signs of disease (Fig 1E).

We then repeated the experiment described in Fig 1B and tested for the presence of tumor-specific $CD8^+$ T memory cells, exploiting luciferase as surrogate tumor-associated antigen (Limberis *et al*, 2009). We recovered splenocytes from 4 groups of mice: (i) naïve (not injected with MM cells, not treated), (ii) injected with MM cells and surviving after being treated with control (PBS), (iii) injected with MM cells and sacrificed because of tumor progression despite being treated with BoxA (BoxA not cured), and (iv) injected with MM cells and surviving after being treated with BoxA (BoxA cured). The splenocytes were cultured for 5 days in the presence of the luciferase peptide GFQSMYTFV to expand LUC-specific T cells and then stimulated (pulsed) or not (unpulsed) for 4 h with the Luc peptide. Splenocytes from BoxA-cured mice contained a significantly higher percentage of $CD8^+CD44^{high}$ IFNγ-producing T cells upon peptide stimulation than splenocytes from mice of the other treatment groups (Fig 1F and G; $P < 0.005$, Kruskal–Wallis test). This experiment indicates that BoxA promotes T-cell responses to a surrogate tumor-associated antigen. In spontaneously surviving mice (not treated with BoxA), the absence of detectable populations of LUC-specific T cells may reflect sub-optimal priming and a relative paucity of antitumor T-cell clones.

We also tested the requirement for T cells in BoxA-dependent antitumor responses. We depleted mice of $CD8^+$ T cells (Appendix Fig S1) prior to the inoculation of MM cells, and then we treated them with either BoxA or PBS (Fig 2A). All CD8-depleted mice developed tumors regardless of treatment (BoxA or PBS) and were sacrificed after 2 weeks. In contrast, some of the non-depleted control mice survived longer than their control counterpart, and this was further promoted by BoxA (Fig 2B and C).

Overall our results show that transplantation of MM cells evokes a spontaneous protective immune response, which can lead to tumor rejection and immunological memory in a small fraction of mice. BoxA boosts immune-mediated recognition, increases the number of

T cells that recognize tumor-associated antigens, and increases the fraction of mice that develop long-term antitumor immunity.

## BoxA induces the relocation of calreticulin without causing cell death

We had expected that BoxA might interfere with ICD, but the results reported in the previous section showed that BoxA favors antitumor immune responses and immunization. We then investigated whether BoxA might instead promote ICD.

BoxA induced an increase in the surface exposure of calreticulin to an extent comparable to the well-known ICD inducer mitoxantrone (MTX) (Figs 3A and EV2A). Tumor masses explanted from mice treated with BoxA displayed calreticulin on the surface of cells, contrary to tumors from untreated mice (Fig 3B). BoxA also induced the release of HMGB1 (Fig 3C), although less efficiently than MTX, and the phosphorylation of eIF2α (Fig 3D), which is pathognomonic for ICD (Bezu *et al*, 2018). However, we detected a transient and dose-dependent activation of each of the three branches of the UPR (Fig EV2B–D), whereas MTX activated only the PERK-eIF2α arm (Fig 3D), in line with data from the literature (Panaretakis *et al*, 2009).

BoxA inhibited MM cell proliferation (Fig 3E); however, it induced no apoptosis, detected as caspase-3 cleavage (Fig 3F and G). Mice engrafted with MM cells that were pretreated or not with 800 nM BoxA for 16 h had comparable tumor growth and overlapping survival curves (Fig 3H and I), which is in keeping with the fact that BoxA does not induce MM cell death. In stark contrast, tumor cells treated with classical ICD inducers before inoculation confer antitumor immunization (Apetoh *et al*, 2007).

These data show that BoxA has a direct antiproliferative effect on tumor cells, where it induces stress and release of DAMPs, but does not cause apoptotic cell death, which is a hallmark of ICD. We thus inferred that tumor eradication by BoxA exploits a pathway different from ICD.

## BoxA promotes tumor cell phagocytosis by macrophages

We showed that BoxA induces cell stress but no apoptosis; thus, we investigated a non-cell autonomous death mechanism that would provide tumor antigens for cross-presentation. Macrophages play a significant role in recognition and clearance of foreign, aged, and damaged cells, and their role in immunosurveillance has been reported (Jaiswal *et al*, 2009; Willingham *et al*, 2012). We therefore tested whether macrophages can ingest MM cells treated with BoxA.

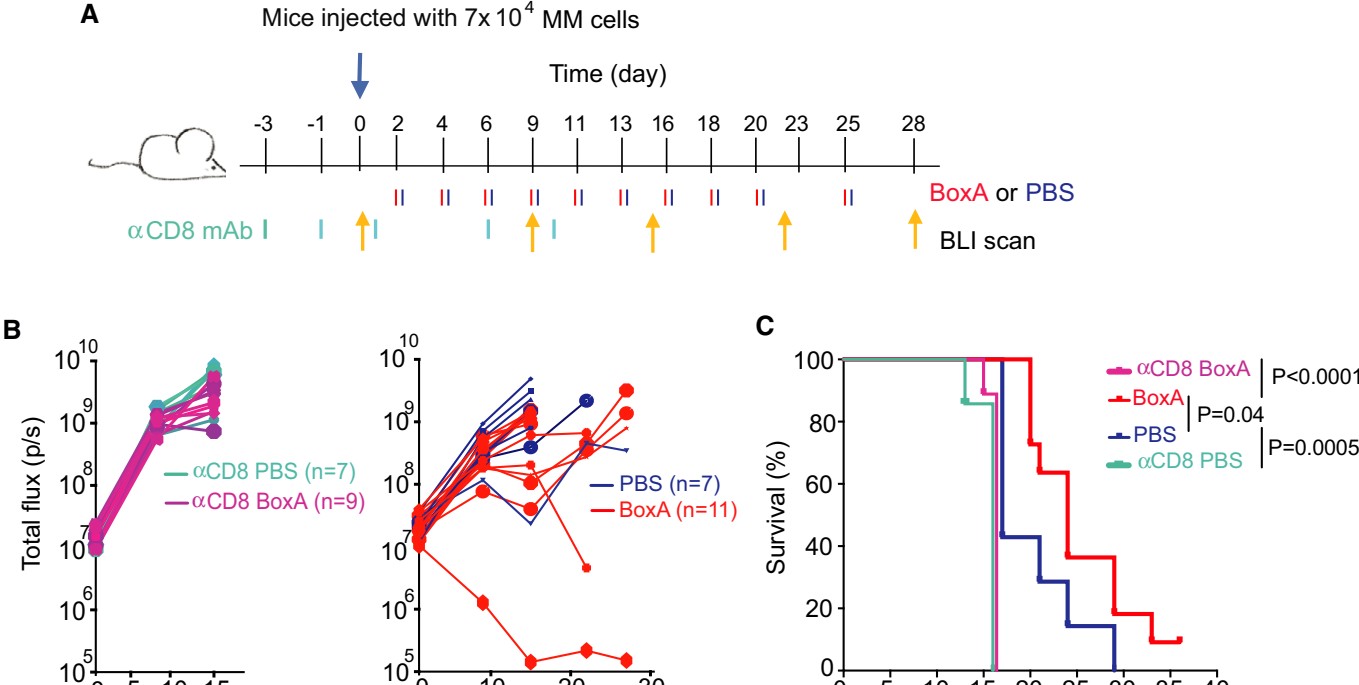

**Figure 2. BoxA exerts its therapeutic effect through CD8⁺ T cells.**

A   Scheme of the experiment. BALB/c mice were depleted or not of CD8⁺ T cells 3 days before the injection of $7 \times 10^4$ MM cells and subsequently treated with either 800 µg BoxA ($n = 9$) or PBS ($n = 7$) three times a week, for 10 times in total. Every week mice were surveyed by BLI.

B   Representation of the BLI of the four groups of mice. Left panel: CD8⁺ T-cell depleted mice treated with BoxA ($n = 9$) or PBS ($n = 7$). Right panel: non-depleted mice treated with BoxA ($n = 11$) or PBS ($n = 7$).

C   Kaplan–Meier survival curves of the 4 groups of mice shown in panel (B). Statistics: Gehan–Breslow–Wilcoxon test.

Source data are available online for this figure.

Indeed, time-lapse imaging (Fig 4A and Movie EV1) of MM cells expressing GFP (GFP+ MM cells) co-incubated with bone marrow-derived macrophages indicated that BoxA increased the engulfment of tumor cells in a dose-dependent manner, from < 1% to almost 20% (Fig 4B).

Phagocytosis by macrophages depends on the balance of "eat me" signals, such as calreticulin, and "don't eat me" signals, such as CD47. A decrease of about one-third in surface CD47 was previously reported to be effective in promoting tumor cell phagocytosis

**Figure 3. BoxA induces the release of damage-associated molecular pattern molecules (DAMPs) but not apoptosis in MM cells.**

A   Surface calreticulin on MM cells was evaluated by flow cytometry after exposure to 800 nM BoxA or 1 µM MTX for 4 or 6 h. Statistics: One-way ANOVA with Dunnett's post-test; $n = 3$.

B   Representative immunohistochemical staining for calreticulin in tumor masses of control (PBS) and BoxA-treated mice. Scale bar 50 µm. The distribution of calreticulin-positive pixels describes the location of calreticulin (intracellular versus ecto-calreticulin). Statistics: Mann–Whitney test, $n = 4$.

C   HMGB1 levels in the medium of MM cells treated with BoxA or 1 µM MTX for 24 h. Statistics: One-way ANOVA with Dunnett's post-test; $n = 3$.

D   Ratio of phosphorylated eIF2α (p-eIF2α) over eIF2α in MM cells treated with increasing concentrations of BoxA or 1 µM MTX for 24 h; β-actin is shown as loading control. Statistics: One-way ANOVA with Dunnett's post-test; $n = 2$.

E   MM cells ($3 \times 10^5$) were grown for 16 h before adding increasing concentrations of BoxA or 1 µM MTX and then further grown for 24 h; t-16 h indicates the number of cells at the start of the experiment.

F   Western blot analysis of cleaved caspase-3 in MM cells exposed or not to BoxA or MTX for 24 h; α-tubulin was used for normalization. Statistics: One-way ANOVA with Dunnett's post-test; $n = 2$.

G   Confocal immunofluorescence microscopy of cleaved caspase 3 (red) in MM cells treated for 24 h with MTX (1 µM) or increasing concentrations of BoxA. Nuclei were stained with Hoechst (blue) and cytosol with phalloidin (green). Scale bar 50 µm.

H   Tumor growth detected via BLI. MM cells pretreated or not with 800 nM BoxA overnight were injected in mice (PBS group, $n = 7$; BoxA pretreated, $n = 6$).

I   Kaplan–Meier survival curve of mice injected with MM cells pretreated with PBS or BoxA shown in panel (H) (Log-rank Gehan–Breslow–Wilcoxon test, $P = 0.59$).

Data information: In panels (A, C, D, E, F), columns represent the average and bars standard deviation of a representative experiment out of at least 2 performed.
*$P < 0.05$, **$P < 0.01$, ***$P < 0.001$, ****$P < 0.0001$.
Source data are available online for this figure.

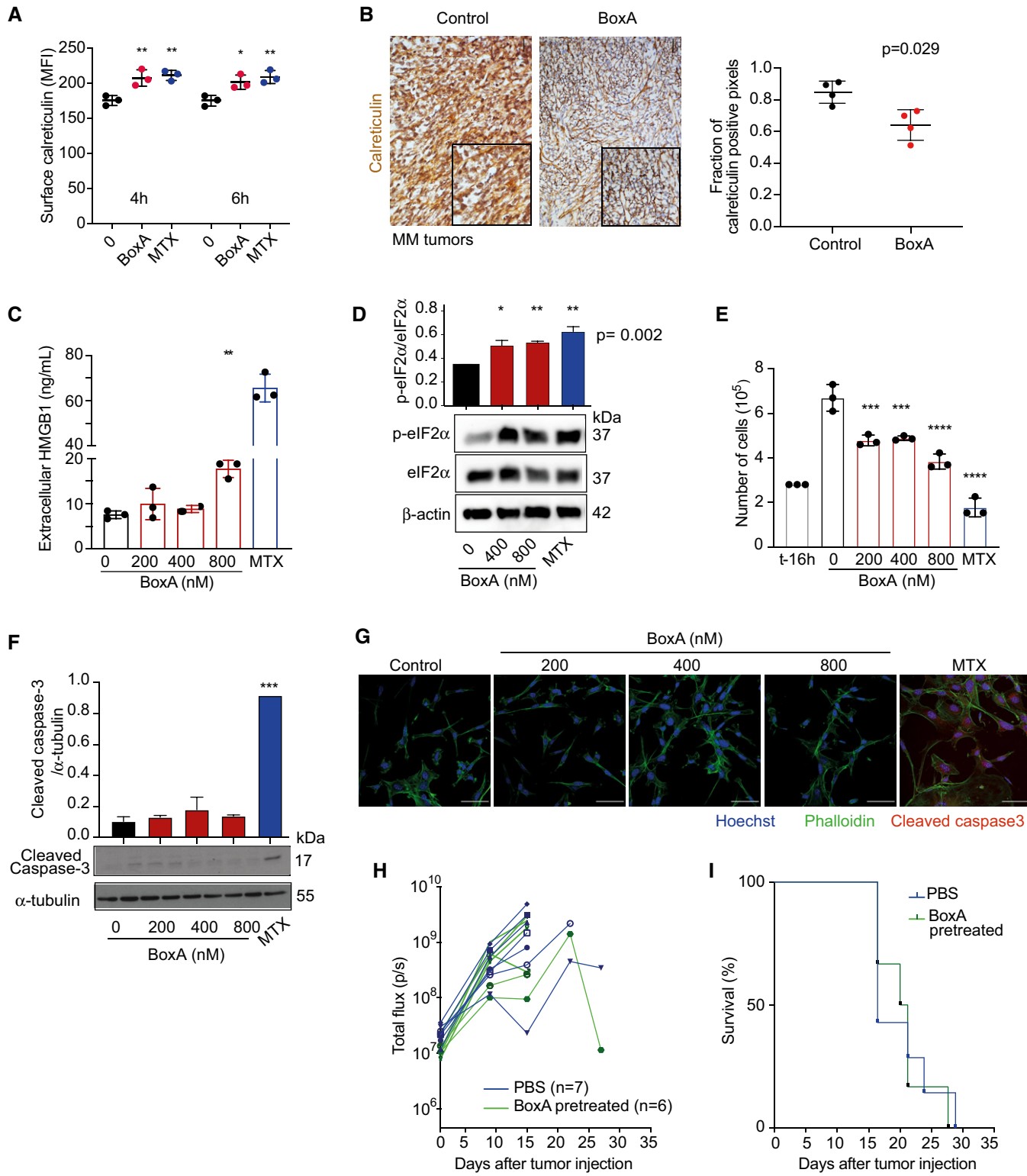

**Figure 3.**

by macrophages (Chao *et al*, 2010). Indeed, flow cytometry indicated that MM cells do express CD47 and that increasing concentrations of BoxA significantly reduced its surface exposure after overnight treatment (Fig 4C and D). CD47 was internalized (Fig 4E,

immunofluorescence imaging) and did not appear to be degraded or shed into the medium (Fig EV3A and B).

Together, these results indicate that BoxA induces the internalization of CD47 and thus the unbalancing of "don't eat me" and

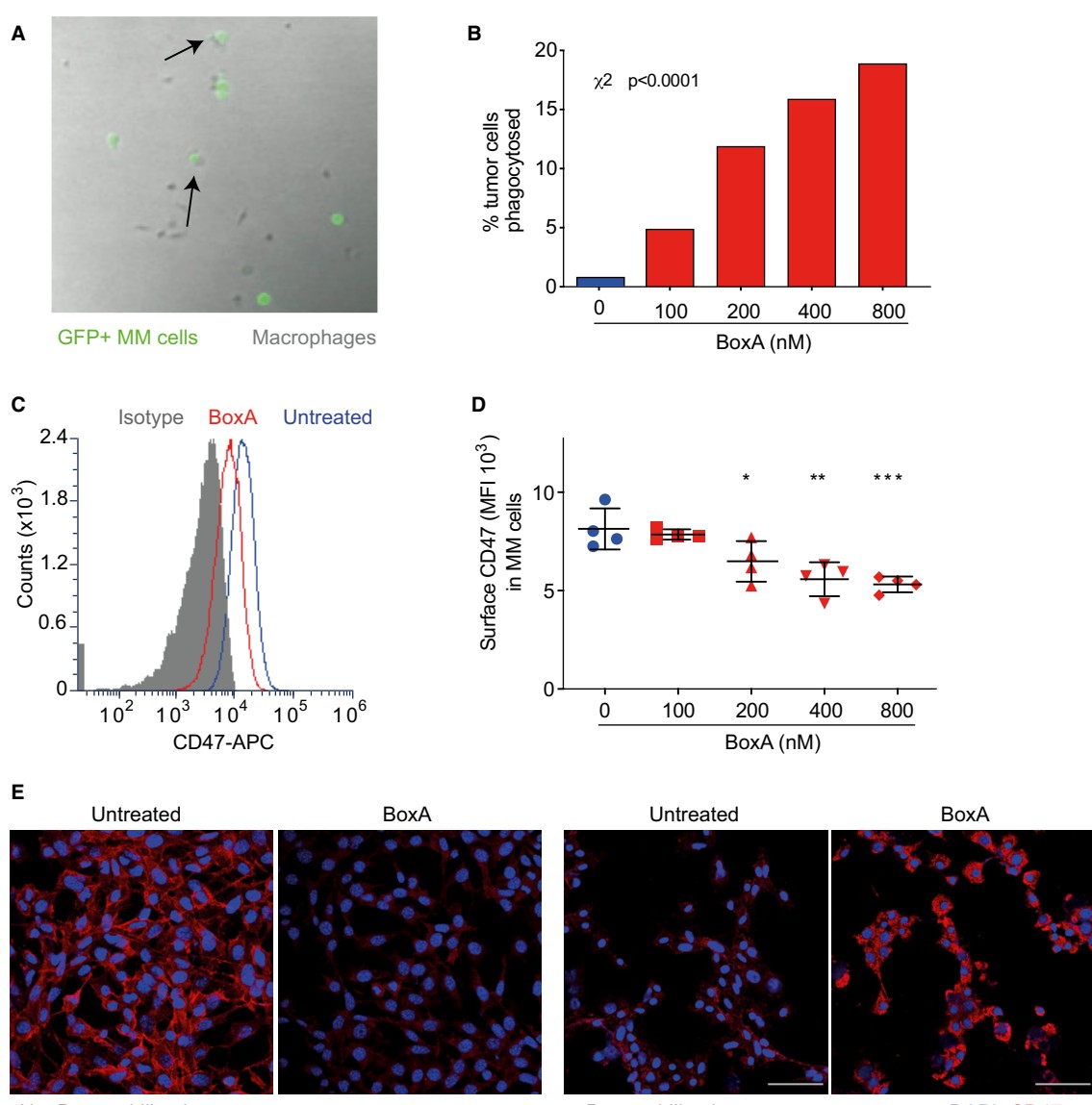

**Figure 4. MM cells exposed to BoxA are phagocytosed by macrophages following CD47 internalization.**

A   Representative frame of a 16-h time-lapse microscopy experiment; GFP[+] MM cells (green) were co-cultured with mouse bone marrow-derived macrophages (gray) in the presence of 200 nM BoxA. Arrows indicate GFP[+] MM cells engulfed by macrophages. This experiment was repeated twice.
B   Fraction of phagocytosed GFP[+] MM cells after 16 h. Statistics: $\chi^2$ test, phagocytosed versus non-phagocytosed MM cells ($n = 68$ to $158$).
C   Surface CD47 in MM cells, untreated (blue) or treated for 24 h with 800 nM BoxA (red). Control isotype (gray).
D   Surface CD47 on MM cells evaluated by flow cytometry after incubation for 24 h with the indicated concentrations of BoxA. The experiment shown is representative of three performed. Error bars indicate standard deviation. Statistics: One-way ANOVA plus Dunnett's post-test. *$P < 0.05$, **$P < 0.01$, ***$P < 0.001$.
E   Representative immunofluorescence staining of CD47 (red) in unpermeabilized and permeabilized MM cells treated or not with 800 nM BoxA for 24 h. Nuclei are counterstained with DAPI. Scale bar 50 µm.

Source data are available online for this figure.

"eat me" signals, which is sufficient to allow MM cell phagocytosis by macrophages.

**BoxA exerts therapeutic effects in a model of colon cancer**

Given the widespread expression of surface CD47 in tumors (Chao *et al*, 2011), we tested if other tumor cell lines respond to BoxA with a decrease in CD47 surface exposure. Cell lines MC38 (mouse colon cancer), B16 (mouse melanoma) and U87 (human glioblastoma) exposed to 800 nM BoxA also reduced surface CD47 after 24 or 48 h (Fig EV4A). We also extensively tested another human cell line—colorectal adenocarcinoma LoVo—and confirmed that BoxA-induced CD47 surface downregulation is associated with the exposure of calreticulin on the plasma membrane, the release HMGB1 in

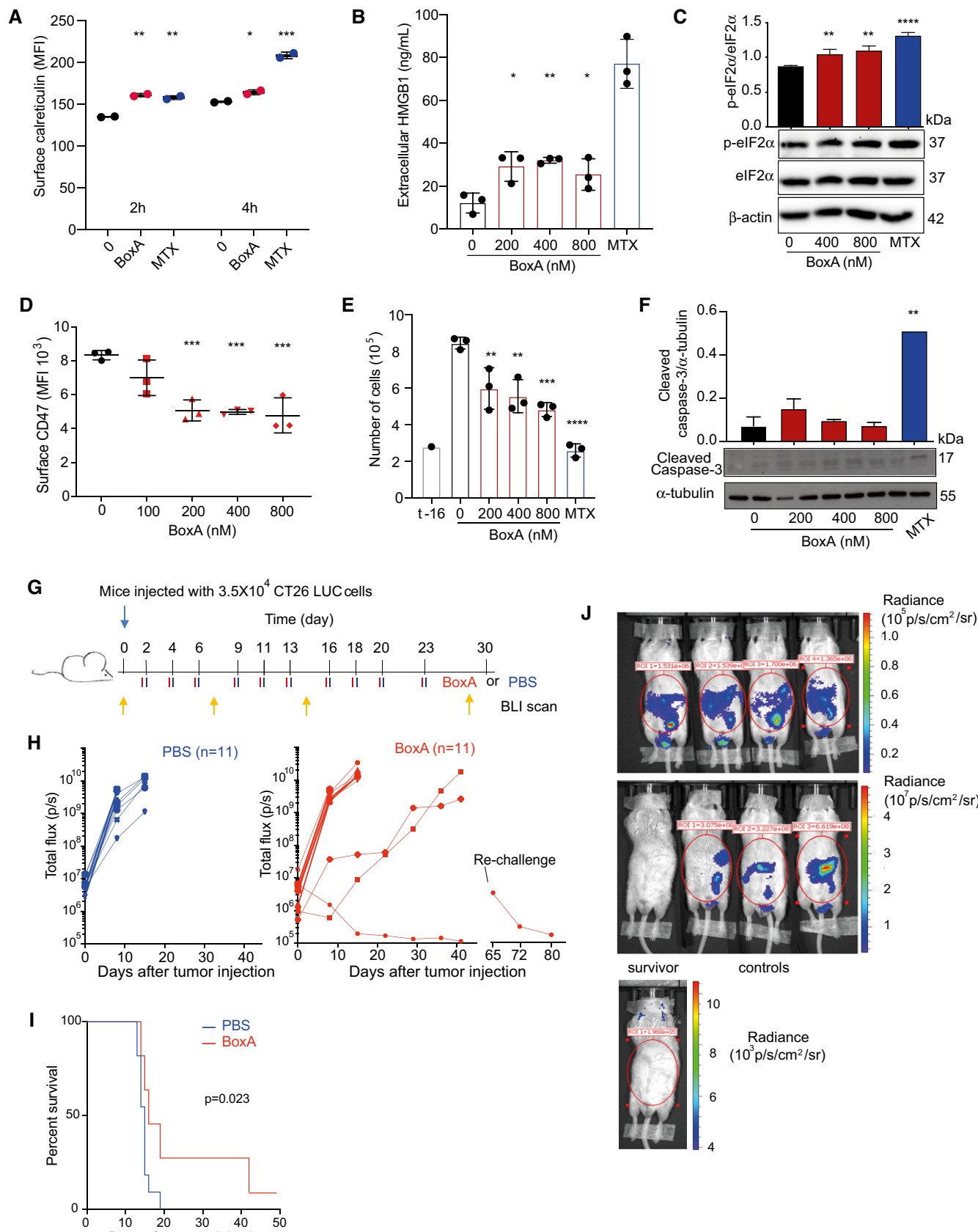

**Figure 5.**

**Figure 5. BoxA exerts therapeutic effects in a model of colon cancer.**

A   Ecto-calreticulin exposure (expressed as MFI) in CT26 cells treated with 800 nM BoxA or 1 μM MTX for 2 and 4 h. Error bars indicate standard deviation. Data were pooled from two different experiments.

B   HMGB1 levels in the medium of CT26 cells treated with BoxA or 1 μM MTX for 24 h ($n = 3$).

C   Ratio of phosphorylated eIF2α (p-eIF2α) over eIF2α in CT26 cells treated with increasing concentrations of BoxA or 1 μM MTX for 24 h. β-actin is shown as loading control ($n = 3$).

D   Surface CD47 on CT26 cells after incubation for 24 h with the indicated concentrations of BoxA. The experiment shown is representative of 2 performed, in biological triplicate.

E   Cell proliferation. CT26 cells ($3 \times 10^5$) were seeded at t-16 h, then exposed to increasing concentrations of BoxA or 1 μM MTX, and counted after further 24 h ($n = 3$).

F   Western blot analysis of cleaved caspase-3 in CT26 cells exposed or not to BoxA or MTX for 24 h. α-tubulin was used for normalization ($n = 2$).

G   Scheme of the experiment. Twenty-two BALB/c mice were inoculated i.p. with $3.5 \times 10^4$ CT26-LUC cells and treated with either 800 μg BoxA or PBS three times a week, 10 times in total. Yellow arrows represent BLI imaging.

H   Tumor growth was detected by BLI. Lines that do not reach day 20 correspond to mice that were sacrificed for ethical reasons. One BoxA-treated mouse survived without any BLI signal and was re-challenged with CT26-LUC cells at day 65; at day 79, it was sacrificed and no tumor masses were detected.

I   Kaplan–Meier survival curves of mice shown in panel (H). Statistics: Gehan–Breslow–Wilcoxon test.

J   BLI was measured 2 h after re-challenge (upper images) and 1 week later (lower images). The survivor was imaged also at much higher sensitivity (note the different scales for radiance).

Data information: In panels (A-F), bars and error bars represent mean ± SD; statistics: One-way ANOVA with Dunnett's post-test, *$P < 0.05$, **$P < 0.01$, ***$P < 0.001$, ****$P < 0.0001$.

Source data are available online for this figure.

the medium, and the activation of the three arms of the UPR (Fig EV4B–E).

We then set up a syngeneic tumor model of colorectal carcinoma (CC), which is among the most common cancers worldwide in terms of incidence and mortality, and is a typical example of an inflammatory tumor. First, we investigated the effect of BoxA on the mouse colon adenocarcinoma cell line CT26: It induced calreticulin exposure (Figs 5A and EV5A), HMGB1 release (Fig 5B), the activation of the three arms of the UPR (Figs 5C and EV5B and C), and the depletion of surface CD47 (Fig 5D), without inducing cell death but merely inhibiting cell growth (Figs 5E and F, and EV5D).

We then injected different numbers ($2 \times 10^4$, $5 \times 10^4$ and $10^5$) of CT26 cells expressing LUC (CT26-LUC) in the peritoneum of BALB/c mice and we followed tumor growth by BLI (Fig EV5E). Most mice died within 16 days, suggesting that this model is far more aggressive compared to the AB1 mesothelioma model. Mice injected with $3.5 \times 10^4$ CT26-LUC cells developed a large number of small tumor masses (around 50) in the abdomen in about 2 weeks (Fig EV5F). These tumors showed very high expression of HMGB1 in the nuclei and in the cytosol and were infiltrated by macrophages and T cells (Fig EV5G).

To test BoxA in the CT26 model, we treated the mice as indicated in Fig 5G. All control mice died within 19 days (Fig 5H); BoxA-treated mice survived significantly longer, and one eventually showed no BLI signal (Fig 5H and I). The surviving mouse was then re-challenged with $3.5 \times 10^4$ CT26-LUC cells, which it rejected (Fig 5H and J); at necroscopy, we could not find any residual tumor mass. In contrast, three control naïve mice injected at the same time developed sizeable tumor masses (Fig 5J) and were sacrificed for severe illness after 14 and 18 days.

**CXCR4 engagement promotes CD47 internalization**

We next investigated which receptor mediates the immunotherapeutic effects of BoxA. BoxA was previously shown to interact with TLR4, RAGE, and CXCR4 receptors (Tirone *et al,* 2018), and MM cells express transcripts for all of them (Fig EV6A). Proximity Ligation Assays (PLA) (Soderberg, 2006) executed on CD47 and CXCR4 gave a strong signal on the plasma membrane of resting MM cells, whereas PLA executed with CD47 and RAGE or CD47 and TLR4 gave 100-fold lower signals (Fig 6A). Pairs of identical receptor molecules were also detected (Fig EV6B), confirming that PLA assays and individual antibodies were functioning properly. Thus, on the surface of unstimulated MM cells a sizeable fraction of CD47 is already in close contact with CXCR4, but not with RAGE or TLR4. Incubation with BoxA caused the almost complete disappearance of

**Figure 6. Engagement of CXCR4 by BoxA mediates CD47 internalization.**

A   Confocal images of Proximity Ligation Assays performed on CD47 and CXCR4, CD47 and RAGE, CD47 and TLR4. Representative images from one experiment out of three performed are shown. MM cells were incubated overnight with 400 nM BoxA or PBS. Nuclei are in blue (DAPI), phalloidin is in green. Red dots represent physical contact of CD47 with CXCR4 or RAGE or TLR4. Scale bar, 20 μm. The intensity of red signal in PLA was quantified as described in Materials and Methods in individual cells ($n$ as indicated). Insets are enlargements of the y axis. Mean and SD are indicated. Statistics: Kolmogorov–Smirnov test.

B   *Cxcr4* mRNA expression in MM cells transfected with sh*Cxcr4* (pools 1 and 2) relative to MM cells transfected with shScramble (AB1 shCTR), one biological replicate, in technical triplicate.

C   Cell proliferation. shCTR ($n = 3$) and sh*Cxcr4* MM cells (pools 1+2, $n = 6$) were plated ($4.4 \times 10^4$ at the start of the experiment, t-16 h), treated or not with 800 nM BoxA for 24 h and counted. Statistics: t-test.

D   Flow cytometry analysis of surface CD47 in shCTR and sh*Cxcr4* MM cells (pools 1+2) exposed or not to 800 nM BoxA for 24 h.

E   Flow cytometry analysis of surface CD47 on MM cells after exposure to 800 nM BoxA and 100 nM AMD3100 for 24 h.

Data information: In panels (D, E) bars and error bars represent mean ± SD; statistics: One-way ANOVA plus Tukey's post-test. In all panels, error bars indicate standard deviation; *$P < 0.05$, **$P < 0.01$, ***$P < 0.001$, ****$P < 0.0001$.

Source data are available online for this figure.

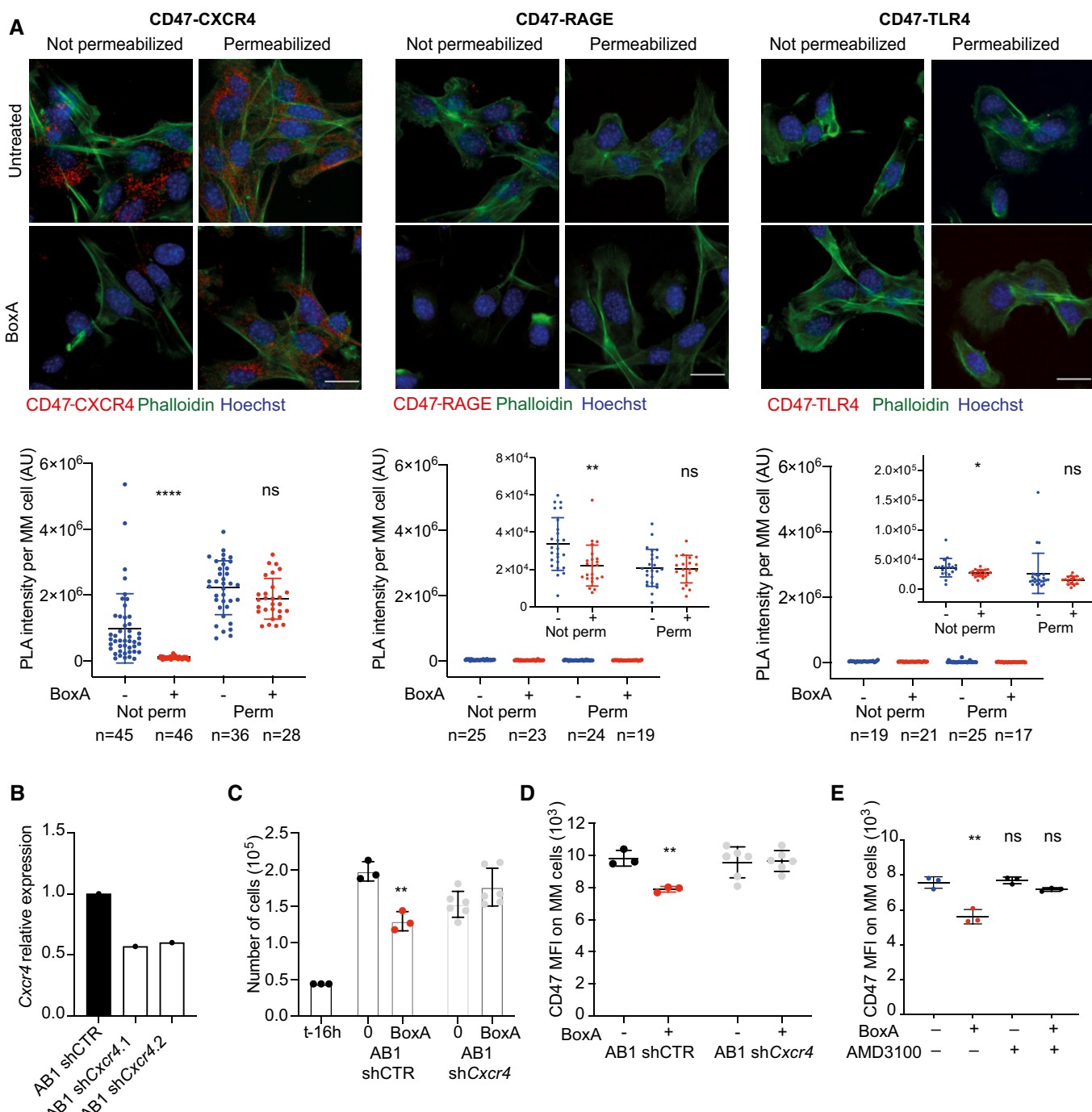

**Figure 6.**

the CD47/CXCR4 PLA signal from the cell surface, whereas it persisted in the cytoplasm (Fig 6A). BoxA also caused a modest decrease of the weak CD47/RAGE and CD47/TLR4 PLA signals on the plasma membrane, consistent with a general decrease of CXCR4 on the cell surface and thus of its availability to contact other surface molecules. These results suggest that CXCR4 cointernalizes with CD47 upon binding BoxA. Indeed, MM cells where CXCR4 expression was reduced by stable silencing (Fig 6B) were not responsive to BoxA: it did not reduce their proliferation (Fig 6C) nor

did it cause CD47 depletion from their surface (Fig 6D). Moreover, AMD3100 (Plerixafor), a specific inhibitor of CXCR4 that hinders its internalization (Hitchinson *et al*, 2017), interfered with BoxA-induced depletion of surface CD47 (Fig 6E).

We then asked whether CXCL12, the natural CXCR4 ligand, would have effects similar to those of BoxA. Indeed, exposure of MM cells to 10–30 nM CXCL12 induced eIF2α phosphorylation (Fig 7A), the release of HMGB1 (Fig 7B), and internalization of surface CD47 (Fig 7C and D). Accordingly, 10 nM CXCL12 also

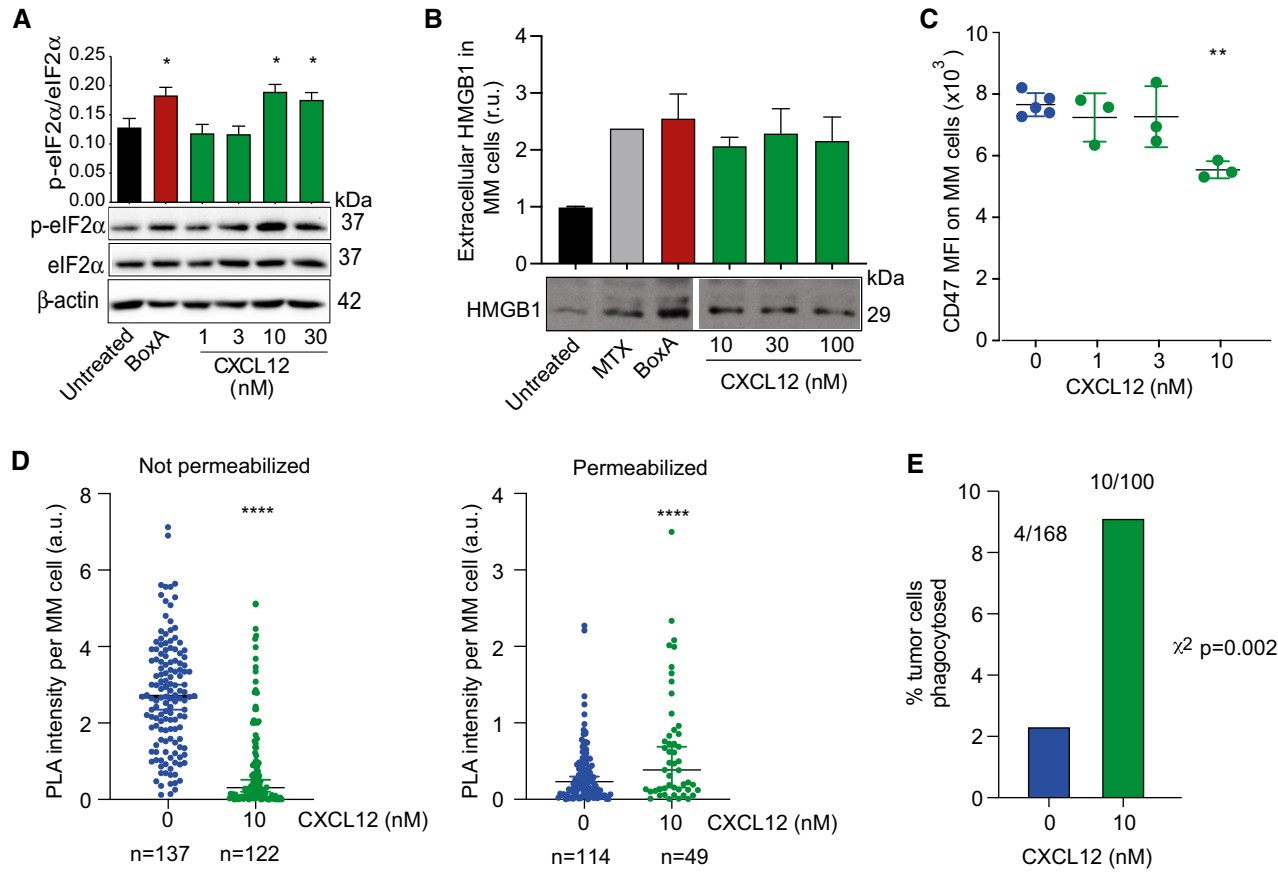

**Figure 7. CXCL12 induces CD47 internalization and macrophage phagocytosis.**

A   Ratio of phosphorylated eIF2α (p-eIF2α) over eIF2α in MM cells exposed to 800 nM BoxA or increasing concentrations of CXCL12 for 24 h ($n = 2$). β-actin is shown as loading control.

B   Western blot of HMGB1 released from MM cells exposed to BoxA and CXCL12 for 24 h. HMGB1 is expressed as relative units compared to untreated cells. The samples ($n = 2$) were loaded on the same gel, but intervening non-relevant samples were cropped from the image. Ponceau S staining was used as a loading control.

C   Surface CD47 on MM cells after incubation for 24 h with the indicated concentrations of CXCL12. The experiment shown is representative of 2 performed, in biological triplicate.

D   PLA intensity was quantified as described in Materials and Methods in individual cells ($n$ as indicated). Mean and SD are indicated. Statistics: Kolmogorov–Smirnov test.

E   Fraction of phagocytosed GFP$^+$ MM cells after 16-h treatment with 10 nM CXCL12. The numbers above the columns indicate the phagocytosed/non-phagocytosed MM. Statistics: $\chi^2$ test, phagocytosed versus non-phagocytosed tumor cells.

Data information: In panels (A-C), bars and error bars represent mean ± SD; statistics: one-way ANOVA plus Dunnett's post-test. In all panels, *$P < 0.05$, **$P < 0.01$, ****$P < 0.0001$.

Source data are available online for this figure.

caused tumor cell phagocytosis in co-cultures of MM cells and macrophages (Fig 7E).

Overall, these results show that CXCR4 engagement, whether triggered by BoxA or CXCL12, causes its co-internalization with CD47.

## Discussion

In this study, we demonstrate that the engagement on the surface of tumor cells of the CXCR4 receptor induces the depletion of surface CD47, the molecule that physiologically prevents phagocytosis by macrophages. As a consequence of CXCR4 engagement, in two

different tumor models, mice develop an antitumor immune response that depends on CD8$^+$ T cells and acquire antitumor immune memory. We designate this mechanism "Immunogenic Surrender" because the tumor cells surrender to macrophages in response to CXCR4 activation, which generally promotes their growth and dissemination.

We performed most of our experiments on a mouse mesothelioma model using BoxA, a fragment of HMGB1. We found that BoxA triggers the activation of the three arms of the UPR, the release of HMGB1, the exposure of the "eat me" signal ecto-calreticulin and the internalization of the "don't eat me" signal CD47. Several of these events and molecular actors are also associated with immunogenic cell death; notably, both ICD and Immunogenic

Surrender cause the release of DAMPs from tumor cells. However, Immunogenic Surrender activates all branches of the UPR, whereas ICD activates one (Panaretakis *et al*, 2009) and does not promote apoptosis, which is functionally replaced by cell phagocytosis via depletion of surface CD47.

We envision Immunogenic Surrender as an antitumor immuno-surveillance mechanism that occurs spontaneously as consequence of CXCR4 activation by CXCL12 or by the HMGB1•CXCL12 hetero-complex. Indeed, in our MM model a small number of mice sponta-neously reject MM cells and develop antitumor immunity. In fact, we found that CD47 internalization is also induced upon binding of CXCR4 by CXCL12. It is no coincidence that BoxA, the molecule that we found to induce Immunogenic Surrender, is a fragment of HMGB1 and behaves in part like HMGB1 and its heterocomplex with CXCL12.

A role of CXCR4 in antitumor responses is unexpected, since the CXCL12-CXCR4 axis has been so far correlated with tumor initiation and progression (Guo *et al*, 2016). Indeed, both HMGB1 and CXCL12 promote tumor cell growth, survival, and invasion (Teicher & Fricker, 2010; Jube *et al*, 2012). In fact, the HMGB1-CXCL12-CXCR4 axis is involved in tissue repair and regrowth after damage (Tirone *et al*, 2018) and may be promoting tumor cell growth in similar ways (Bianchi & Mezzapelle, 2020). In this context, Immunogenic Surrender can be seen as a counterbalancing activity of the HMGB1-CXCL12-CXCR4 axis, one that induces the repairing/growing tissue to submit itself to immunological scrutiny. We speculate that such a check-and-balance should provide a definite evolutionary advantage.

We suggest that several tumor types activate Immunogenic Surren-der during their development. We verified that BoxA promotes UPR, DAMPs release, and CD47 surface depletion in several mouse and human tumor cell lines; most importantly, BoxA induces tumor rejec-tion and immunization also in mice inoculated with colon carcinoma CT26 cells. We note that BoxA has immunotherapeutic effects similar to the blockade of CD47 with monoclonal antibodies; in fact, antibody targeting of CD47 can be seen as an extrinsic therapeutic intervention that mimics and exploits the CD47 surface depletion that occurs in Immunogenic Surrender.

Critically, in our work we unveiled Immunogenic Surrender using BoxA, which acts as an antagonist on TLR4 and RAGE, and is a partial activator of CXCR4. BoxA suppresses tumor cell growth while, in contrast, CXCL12 promotes tumor cell growth (Guo *et al*, 2016). These contrasting effects might be due to the remarkable plas-ticity of G-coupled protein receptors in the activation of downstream signaling pathways after binding to different yet related agonists—a process called biased signaling (Violin *et al*, 2014; Hitchinson *et al*, 2017). The difference in cell growth induced by BoxA and CXCL12 is critical from a translational point of view: Even if both were equally potent in inducing Immunogenic Surrender, injecting CXCL12 would also promote cancer growth, while our data indicate that injecting BoxA does restrain it. BoxA is not toxic to mice, possibly because it may be more effective on tumor cells than on normal cells. In the perspective of drug development, BoxA can be considered a hit molecule that enabled the discovery of Immunogenic Surrender; such a hit might be improved by design or entirely replaced by small molecules that bind to CXCR4 and promote CD47 internalization, while not retaining the full agonist activity of CXCL12.

In conclusion, we have unveiled a previously unknown activity of CXCR4, which by co-internalizing CD47 exposes tumor cells to immunosurveillance. We have also identified a molecule, BoxA, which is not cytotoxic but can enhance the immunosurveillance-related activity of CXCR4 without promoting cell growth. We suggest that BoxA-instructed recognition of tumor antigens should be comple-mentary to checkpoint inhibitors, which potentiate the response of immune cells toward already recognizable tumor antigens. The use of BoxA, or a small molecule mimicking its action on CXCR4, may hold promise for malignant mesothelioma, for which there are very few available therapeutic interventions, and more generally for the wide variety of tumors vulnerable to CD47 blockade.

# Materials and Methods

### Cell lines and drug compounds

AB1 mouse mesothelioma cells (Cell Bank Australia), CT26 mouse colon cancer cells, and LoVo human colon cancer cells (both obtained from ATCC, Manassas, VA, USA) were cultured at 37°C under 5% $CO_2$ in RPMI 1640 (Life Technologies) supplemented with 5 or 10% v/v fetal bovine serum (Life Technologies), respectively, 2 mM L-glutamine, 100 U/ml penicillin/streptomycin, and 10 mM HEPES. Luciferase-expressing AB1-B/c-LUC cells were previously described (Mezzapelle *et al*, 2016). CT26-LUC cells were obtained by transfecting CT26 cells with the pLenti PGK V5-LUC Neo (w623-2) plasmid (Addgene #21471). MC38 cells were kindly provided by Mario Colombo (Fondazione IRCCS Istituto Nazionale dei Tumori, Milano, Italy), B16 cells by Matteo Bellone and U87 cells by Andrea Graziani (both IRCCS Ospedale San Raffaele, Milano).

All cell lines were passaged for no longer than 10 passages after thawing. Cell lines were routinely tested for mycoplasma contami-nation by PCR. To elicit the endoplasmic reticulum (ER) stress, cells were treated 1 µM Mitoxantrone (MTX, Sigma-Aldrich) for 24 and 48 h. AMD3100 was purchased from Sigma-Aldrich, BoxA, and CXCL12 (LPS-free) from HMGBiotech (Milano, Italy).

### Mice

Animal experiments were approved by the Animal Care and Use Committee (IACUC #839) of Ospedale San Raffaele in accordance with the Italian law. Eight-week-old male BALB/c mice were purchased from Charles River Laboratories (Calco, Italy). Animals were housed under specific pathogen-free conditions and allowed access to food and water ad libitum. Mice were inoculated intraperitoneally (i.p.) with $7 \times 10^4$ MM or with $3.5 \times 10^4$ CT26-LUC cells. Cell engraftment was confirmed by bioluminescence imaging (BLI). Two days after inoculation, mice were randomized into experimental groups and treatments started. BoxA (800 µg) was administered i.p. three times a week for 3 weeks (10 times in total). Control mice were injected i.p. with saline. Tumor growth was assessed weekly by BLI. Mice were monitored daily and were sacrificed when severely distressed (BLI signal $> 10^9$ photons/s).

### *In vivo* BioLuminescence optical imaging (BLI)

BLI was performed on mice after the i.p. injection of $7 \times 10^4$ MM cells or $3.5 \times 10^4$ CT26-LUC using an IVIS SpectrumCT Preclinical *In Vivo* Imaging System (Perkin Elmer). The system is equipped with a low noise, back-thinned, back-illuminated CCD camera cooled at −90°C

(quantum efficiency in the visible range above 85%). Before BLI, each mouse received an intra-peritoneal injection of 6 g luciferin/kg body weight. During image acquisition, the animals were kept at 37°C and under gaseous anesthesia (2–3% isoflurane and 1 lt/min $O_2$). After luciferin injection, dynamic BLI was performed from 0 to 30 min by acquiring an image every 2 min (exposure time = auto, binning = 8, f = 1 and a field of view equal to 13 cm (field C)) in order to detect the highest BLI signal. BLI image analysis was performed by measuring the total light flux (photons/s) in a uniform region of interest (ROI) placed over the animal abdomen. Images were acquired and analyzed using Living Image 4.4 (Perkin Elmer).

### Tumor re-challenge

BALB/c mice that rejected the first tumor challenge and naïve BALB/c mice were inoculated i.p. with $7 \times 10^4$ MM cells for the mesothelioma model and with $3.5 \times 10^4$ CT26-LUC cells for the colon carcinoma model. Tumor growth was followed by BLI. Mice were sacrificed when the bioluminescent signal remained at background levels for three consecutive scans.

### Histology and immunofluorescence

#### Mouse samples

Tumor masses were explanted from each sacrificed mouse, fixed in zinc formalin for 24 h, processed with Leica TP1020, embedded in paraffin, and slices were cut. Briefly, the sections (3 μm) were deparaffinized in xylene and rehydrated in graded alcohol. Sections were stained with hematoxylin and eosin. Immunohistochemical staining was performed using the following antibodies: anti-F4/80 (clone A3-1 Bio-Rad), anti-CD3 (clone SP7 Abcam), anti CD45R/B220 (clone RA3-6B2 BD Biosciences), anti-HMGB1 (#18256 Abcam), and anti-calreticulin (#2907 Abcam). Slides were counterstained with hematoxylin and mounted. For immunofluorescent staining, 6-μm-thick serial cryostat sections from mice tumor samples, immediately snap-frozen in OCT after removal, were fixed with cold acetone for 10 min and co-stained with the following Ab: rabbit anti-mouse mAb to F4/80 (1:100, Biolegend), rat anti-mouse mAb to CD206 (1:200 Bio-Rad), and rat anti-mouse mAb to CD86 (clone PO.3, 1:100, #04-1527 Millipore). Alexa Fluor 546 goat anti-rabbit IgG (Molecular Probes) and Alexa Fluor 488 goat anti-rat IgG (Molecular Probes) were used as secondary antibodies. Images were analyzed by confocal microscopy and acquired with Fluoview FV500 software (Olympus BioSystems). For CD86 detection, immunostaining was performed using a streptavidin-biotin-alkaline-phosphatase-complex staining kit (Bio-Spa Division) and naphthol-AS-MX-phosphate and Fast-Red TR (Sigma-Aldrich) to visualize binding sites. Mayer's hematoxylin was used as a counter-staining, followed by mounting in glycergel (Dako). Images were acquired with Leica DM RX microscopy using Scion Image software.

For cleaved caspase-3 immunofluorescence in MM and CT26 cells, $4 \times 10^4$ cells were seeded on coverslips. After 16 h, cells were treated or not with BoxA (200, 400, and 800) nM and MTX 1 μM for 24 h. After treatment cells were fixed with 4% PFA, permeabilized (HEPES-Triton X-100 buffer), and washed three times with PBS 0.2% BSA. After blocking with PBS 4% BSA, cells were stained overnight with anti-cleaved caspase-3 primary antibody (1:800 5A1E, Cell signaling). Cells were then washed three times and labeled with goat anti-rabbit RFP 633 nm (1:1,000, Invitrogen

#A21070) and incubated for 45 min at RT. Phalloidin 594 nm (1:1,000, P5282 Sigma-Aldrich) was used to stain the cytosol and Hoechst 33358 (10 min at RT in PBS) was used to stain the nuclei. The slides were acquired with a 63× objective using a confocal microscope (TCS SP5 AOBS Leica LSM; Leica Microsystems).

For CD47 immunofluorescence in MM cells, $4 \times 10^4$ cells were seeded on coverslips (previously incubated with polylysine for 30 min at 37°C). After 24 h, cells were treated with BoxA 400 and 800 nM for 24 h, washed with 0.2% BSA in PBS, and stained with anti-CD47 antibody (1:100 miap301, BE0270, Bioxcell) for 1 h at 37°C. After labeling, cells were washed three times, fixed with PFA 4% (10 min at RT), and labeled with goat anti-rat Alexa Flour 546 secondary antibody (#A-11081, Invitrogen) for 45 min. Hoechst 33358 (10 min at RT in PBS) was used to stain the nuclei. The slides were acquired with a 63× objective using a confocal microscope (TCS SP5 AOBS Leica LSM; Leica Microsystems).

#### Patients' samples

The study involved mesothelioma patients admitted to the Thoracic Surgery Unit of San Raffaele Hospital (Milano, Italy) and Maggiore della Carità Hospital (Novara, Italy) between 2015 and 2017. Diagnosis of mesothelioma was based on standard histological and immunohistochemical criteria, including positivity to calretinin, vimentin, and cytokeratins 5 and 6, and negativity to carcinoembryonic antigen, thyroid transcription factor 1, and Ber Epy 4. Pleural biopsies were collected and fixed in formalin for 24 h, embedded in paraffin, and processed.

#### Image acquisition and analysis

All images were scanned using the Aperio Scanscope C2 system (Leica Biosystems).

### RNA extraction and real-time PCR analysis

Total RNA was extracted from cells and tumors using NucleoSpin RNA (Macherey-Nagel) and treated with DNase I. The amount of total RNA was determined by UV spectrophotometry using a Nano-Drop Spectrophotometer (Thermo Fisher Scientific). Next, 1 μg of total RNA was reversed transcribed using the Superscript III Reverse Transcriptase (Thermo Fisher Scientific) following the manufacturer's protocol. PCR analysis was carried out using AmpliTaq Gold® DNA Polymerase (Thermo Fischer Scientific).

Receptor expression was evaluated using the following primers:

CXCR4 Forward: 5′ TAGAGCGAGTGTTGCCATGG 3′; Reverse 5′ TGAAGTAGATGGTGGGCAGG 3′.

RAGE Forward 5′ TCCTCAGGTCCACTGGATAAA G 3′; Reverse 5′ TGTGACCCTGATGCTGACAGG 3′.

TLR4 Forward 5′ CAGTGGTCAGTGTGATTGTGG 3′; Reverse 5′ TTCCTGGATGATGTTGGCAGC 3′.

β-actin Forward: 5′AGA CGG GGT CAC CCA CAC TGT GCC CAT CTA 3′; Reverse 5′ CTA GAA GCA CTT GCG GTG CAC GAT GGA GGG 3′.

### Western blotting

Protein extracts from different cell lines were prepared as follows. Cells were lysed by an ice-cold lysis buffer containing 50 mM

Tris–HCl, pH 7.5, 150 mM NaCl, 10 mM EDTA, 1% NP-40, 0.1% SDS supplemented with a cocktail of protease inhibitors (Roche), 1 mM PMSF, 1 mM NaF, and PhosSTOP Phosphatase inhibitor Cocktail (Roche). Samples were sonicated prior to mixing with reducing SDS–PAGE sample buffer and heating (5 min, 90°C). After transferring proteins to PVDF or nitrocellulose, the membrane was blocked with 5% BSA or 5% non-fat dry milk. Antibodies to eIF2$\alpha$ (#9722), p-eIF2$\alpha$ (#3597), and cleaved caspase-3 (#9664) were from Cell Signaling Technology, ATF6 (ab37149), IRE1 (ab37073), HMGB1 (ab18256), and CD47 (ab215616) were from Abcam; $\beta$-actin (F-3022) and $\alpha$-tubulin (T-5168) from Sigma-Aldrich. Anti-mouse or anti-rabbit antibodies, conjugated with horseradish peroxidase, were used as secondary antibodies. An ECL chromogenic substrate was used to visualize the bands. In some experiments, Ponceau S staining was used for assessing equivalent protein loading. To detect extracellular HMGB1, the culture medium was collected and concentrated via centrifugation (5,000 $g$-force for 1 h) through microconcentrators (Centricon plus 10 KDa filter Amicon Ultra, Millipore). Proteins were then analyzed by Western blotting.

Some blots were cut and probed with different antibodies for different proteins, including $\beta$-actin. In some cases, to examine proteins of similar molecular weight, the PVDF membranes were subjected to a mild stripping protocol, as recommended by Abcam. Western blot bands were quantified using Fiji or Image Lab software.

### HMGB1 ELISA

The HMGB1 ELISA Kit (IBL International-Tecan) was used to measure the levels of HMGB1 in the media of AB1 and CT26 cells. Samples were tested in duplicate. $3 \times 10^5$ cells were cultured in 6-well plates in RPMI 5% FBS. After 16 h, fresh media (RPMI 1% FBS) and treatment (0, 200, 400, 800 nM BoxA and 1 $\mu$M MTX) were added for 24 h. The culture media were then collected and concentrated by ultrafiltration using Amicon Ultra Centrifugal Filters (Millipore), and 10 $\mu$l aliquots were assayed in duplicate by ELISA. All culture media were collected under identical condition.

### Cell proliferation

$3 \times 10^5$ AB1 or CT26 cells were seeded in 6-well plates, grown overnight, and treated with BoxA (200, 400, 800 nM) or 1 $\mu$M MTX for 24 h. Each condition was tested in triplicate. After PBS washing, cells were harvested in trypsin-EDTA, centrifuged, and resuspended in 1 ml RPMI supplemented with 5% FCS and counted with a Burker chamber.

### Flow cytometry

Cytofluorometry staining for ecto-calreticulin was performed on both mouse and human fixed cells with a PE-conjugated anti-calreticulin antibody (FMC 75 ab83220). Surface CD47 was detected with a mouse anti-CD47 Alexa 647 antibody (Clone mIAP301, 1:100, BD Biosciences) and the relative control isotype (rat IgG2a,k BD Biosciences) or with anti-human CD47 Alexa 647 or PE (Clone CC2C6, 1:100, BioLegend or B6H12, 1:100, BD Biosciences) and the relative control isotype (mouse IgG1,k BioLegend). Stained cells were analyzed by BD Accuri C6, BD FACS Canto II, or BD FACS Verse.

The efficiency of CD8 T-cell depletion *in vivo* was determined by flow cytometry in peripheral blood at day 5 after MM cell inoculation and in the spleen. Blood and spleen cells were stained with APC-Cy7 conjugated anti-CD3 (1:50, #130-102-306, Miltenyi Biotec) and PerCP-Cy5.5 conjugated anti-CD8 (1:400, clone 53-6.7, BioLegend) antibodies. The stained cells were analyzed by BD FACS Canto II (BD Biosciences). Data were analyzed using the FCS Express 6 software.

### Generation of macrophages

Bone marrow cells were isolated from the femurs of 8-week-old male BALB/c mice. Macrophages were obtained by culturing bone marrow cells in DMEM containing 10% FCS supplemented with macrophage colony-stimulating factor (M-CSF, 20 ng/ml) for 6 days. At day 4, the medium was replaced with fresh M-CSF, and at day 6, adherent macrophages were harvested.

### *In vitro* phagocytosis assay

GFP-positive AB1 mesothelioma cells ($12.5 \times 10^3$) were incubated with $3 \times 10^4$ mouse macrophages in 48-well plates in the presence or not of BoxA (100, 200, 400, and 800 nM) or CXCL12 (10 nM) for 16 h. Cells were observed under a confocal microscope (Leica TCS SP5) using a 20× dry objective (0.7 numerical aperture). The confocal microscope is equipped with an incubation system to maintain the cells at 37°C in 5% $CO_2$. GFP-positive AB1 cells were imaged with low-intensity 488 nm Argon laser, while macrophages were observed in the transmission channel. Images were acquired in 16-bit format ($1,024 \times 1,024$ pixels) every 5 min for 16 h.

### Long-term culture of splenocytes and IFN$\gamma$ intracellular staining

Single-cell suspensions from mouse spleens were obtained by mechanical dissociation; $3 \times 10^7$ splenocytes were cultured for 5 days in RPMI medium plus 10% FCS supplemented with a firefly luciferase-derived peptide (1 $\mu$M, peptide sequence NH2-GFQSMYTFV-COOH, Primm srl) (Limberis *et al*, 2009). After 5 days, vital lymphocytes were isolated using Lympholyte Cell Separation media (Cedralane Labs) and $5 \times 10^5$ lymphocytes were stimulated for 4 h with either $10^6$ RMA cells pulsed (2 mM for 1 h at 37°C, pulsed RMA) or not (unpulsed RMA, as control) with the Luc peptide or with Phorbol Myristate Acetate (PMA)/ionomycin (positive control). Brefeldin A (Sigma-Aldrich) was added to the samples during the last 2 h of culture. Cells were then surface stained for CD8 (PerCP/Cyanine5.5 conjugated, Clone 53-6.7, #100734, Biolegend; 1:200), CD4 (PE-conjugated Clone RM4-5 #100521 Biolegend; 1:100), and CD44 (V450 conjugate, Clone IM7, #560452 BD Biosciences; 1:200), then fixed in 2% paraformaldehyde, permeabilized (0.5% saponin, 2% heat-inactivated FBS, 2% rat serum, 0.2% sodium azide), and further stained for intracellular IFN$\gamma$ Samples were acquired on BD FACS Canto II, and data were analyzed using the FCS Express 6 software.

### *In vivo* CD8 depletion

Two hundred $\mu$g of either anti-mouse CD8 monoclonal antibody (Clone 2.43, BioXcell) or IgG control were injected i.p. at day $-3$,

### The paper explained

**Problem**

Malignant mesothelioma (MM) is a tumor arising from asbestos-induced chronic inflammation and for which few therapeutic options are available. High Mobility Group Box 1 (HMGB1) protein favors the onset and progression of MM. We tested the therapeutic potential of BoxA, a fragment of HMGB1 that competes with the intact protein, in an immune-competent mouse model of MM.

**Results**

We find that BoxA induces MM remission and antitumor immunization in a large fraction of mice. The binding of BoxA to the CXCR4 receptor induces DAMPs release and CD47 internalization, leading to tumor cell phagocytosis by macrophages. CXCL12, the natural ligand of CXCR4, also promotes CD47 internalization.

**Impact**

Our study indicates that the CXCL12/CXCR4 axis, which is known to promote cancer progression, also promotes a counterbalancing antitumor response. BoxA is non-toxic and, contrary to CXCL12, inhibits tumor cell growth. Thus, BoxA, by shifting the balance from tumor growth to antitumor immunization, might hold promise as first-in-class antitumor drug that should be synergic with checkpoint inhibitors. Furthermore, synthetic ligands that act like BoxA may be as effective as anti-CD47 antibodies, which are in advanced clinical development.

−1, +1, +6, +10 relative to the inoculation of MM cells (day 0). CD8 T-cell depletion in the peripheral blood was assessed by flow cytometry and confirmed to exceed 95%. Samples were analyzed on a BD FACS Canto II apparatus and data analyzed using the FCS Express 6 software.

### Proximity ligation assay (PLA)

$2x10^4$ MM cells were seeded on glass coverslips and the following day treated overnight with either BoxA (400 nM) or CXCL12 (10 nM) and PBS (control). Following treatment, cells were fixed with 4% paraformaldehyde in PHEM buffer for 10 min at RT, washed twice with 1% BSA in PBS for 5 min, and then blocked with 4% BSA and 10% goat serum in PBS. Cells were overlayed with the primary antibodies:

rabbit monoclonal anti-CD47 (1:100, EPR21794, Abcam #AB218810); mouse monoclonal anti-CD47 (1:50, B6H12, Santa Cruz #sc12730) either alone or in combination or goat polyclonal anti-CXCR4 (1:100, Abcam #AB1670), or rabbit monoclonal anti-TLR4 (1:50, Cell signaling #14358) and or rabbit polyclonal anti-RAGE (1:100, Invitrogen #PA1-075) for 1 h at room temperature. Following three washes with 0.2% BSA in PBS, the cells were incubated with secondary antibodies in 0.2% BSA/PBS + 10% goat serum and incubated for 45 min at RT. For nuclei staining, 1 μg/ml Hoechst 33358 was used. For cytosol detection, Phalloidin FITC (P5282, Sigma-Aldrich) was used.

Secondary probes (Duolink, Sigma-Aldrich) for PLA reaction were as follows: Anti-Rabbit MINUS (#DUO92005), Anti-Rabbit PLUS (#DUO92002), Anti-Goat MINUS (#DUO92006), and Anti-Goat PLUS (#DUO92003). When both primary antibodies were used, the PLA products were obtained by using the anti-rabbit PLUS and anti-goat MINUS probes.

### PLA quantification

Custom-made MATLAB routines were used to quantify the intensity of the PLA signal for each cell, available upon request. In short, to segment the nuclei, we used the signal from the Hoechst channel. Nuclear masking was performed using as a threshold the mean image intensity plus twice the standard deviation, while segmentation was performed after a watershed transformation allowing to segment most of the overlapping nuclei. The resulting segmented nuclei were filtered by size to exclude artifacts or improperly segmented clusters of nuclei. To estimate the signal intensity per cell, a ring of 100 pixels (8.5 μm) around each segmented nucleus was applied, able to cover most of the cell's surface and of the signal coming from each cell. We then estimated the intensity of the signal per cell as the total intensity of the nucleus plus the intensity of the ring around it, after background subtraction. When two or more rings did overlap in a given region, the intensity of such region was evenly divided between the cells involved.

### Calreticulin IHC quantification

To estimate the calreticulin intensity from immunohistochemistry images, ad hoc MATLAB routines were developed (available upon request). Pixels of the RGB images have three coordinates (corresponding to the red, green, and blue intensities), and we classified them in different "colors" using a K-means clustering algorithm with K = 4 (K > 4 led to similar final results). As a result, this algorithm defines the color of calreticulin-positive pixels. The next step is to measure the distribution of calreticulin-positive pixels: If calreticulin is diffuse within the cytoplasm, a large number of contiguous pixels is positive, and calreticulin-negative pixels are few. Ecto-calreticulin, in contrast, corresponds to a lower number of positive pixels, and a larger number of calreticulin-negative pixels. The fraction of calreticulin-negative pixels, thus, is a descriptor of the location of calreticulin.

### Silencing of CXCR4

AB1 (MM in the text) cells were transfected with a lentiviral expression vector containing either a short hairpin RNA directed against CXCR4 or a scramble control (Open Biosystems code v2MM_217115 SM 2566) using Lipofectamine 3000 (Invitrogen), according to the manufacturer's instructions. Cells surviving puromycin selection were harvested as a pool and analyzed for CXCR4 expression. Total RNA was extracted from AB1 cells shCTR and AB1 sh*CXCR4* (pool 1 and 2) using NucleoSpin RNA (Macherey-Nagel) and treated with DNase I. The RNA amount was determined using the NanoDrop Spectrophotometer (NanoDrop Technology). Next, 1 μg of total RNA was reversed transcribed using the SuperScript III Reverse Transcriptase (Thermo Fisher Scientific) following the manufacturer's protocol. The cDNA reaction was used in triplicate SYBR-Green (Applied Biosystems) qPCR reactions containing primers specific for CXCR4: Forward: 5′-TCA GTG GCT GAC CTC CTC TT-3′; Reverse: 5′-CTT GGC CTC TGA CTG TTG GT-3′. β-actin Forward: 5′AGA CGG GGT CAC CCA CAC TGT GCC CAT CTA 3′; Reverse: 5′CTA GAA GCA CTT GCG GTG CAC GAT GGA GGG 3′. β-actin RNA was used for normalization.

## Statistical analysis

We performed all experiments in duplicate, triplicate, or quadruplicate if we expected that nonparametric comparisons between two groups would be appropriate, in triplicate if we expected that parametric would be appropriate. Most experiments did not have a large enough sample size to test for normal distribution; we used parametric tests only where normality was expected a priori. Experiments that used multiple doses or multiple times were done in duplicate or triplicate. In general, we used the minimum number of replicates that can be analyzed statistically.

For animal studies, the sample size estimate was indicated in the application to obtain an IACUC. In our case, the number of animals was estimated assuming a survival = 0 in the control group and > 30% survival in the treatment group; to get a statistical significance in survival analysis with an 80% power and alpha error for 25%, we needed > 6 animals/group, and we generally used 9 mice/group. In the experiment reported in Fig 1, the sample size was chosen as 20 per group because we expected no surviving animals, but only an extension of their survival; the actual result was unexpected. There were no pre-established criteria for animal exclusion and, in general, we did not exclude animals, save the ones where an error in handling was done, for example, a mouse that died during anesthesia. All animals were assigned a code and assigned to a group after random number generation. The investigator who performed treatments was not blinded, because it either made no sense or was impractical; the investigator who performed BLI measures was a different one and was blinded to which animals were treated or not. For flow cytometry analyses, which often require gating that to some extent is subjective, the technician who collected the data was unaware of which samples were which. Results were analyzed statistically without the option of reallocating or excluding samples.

Statistical analyses were performed with GraphPad Prism software, version 8.1.1 (GraphPad software, Inc.).

## Data availability

This study includes no data deposited in external repositories.

Expanded View for this article is available online.

## Acknowledgements

We thank Eltjona Rrapaj, Emilie Venereau, Alessandra Agresti for scientific discussions, Antonello Spinelli, Laura Perani, and Massimo Venturini for *in vivo* imaging, Amleto Fiocchi for IHC. This work was supported by University of Sannio with grants to VC and LC, by Italiana per la Ricerca sul Cancro (AIRC) with a fellowship to RM and grants to AM (IG 2018-ID21763) and MEB (IG2020-ID27702; IG2019-ID24290; Accelerator award PREDICT-Meso), and by Fondazione Buzzi with a grant to MEB.

## Author contributions

Conceptualization: RM, AM, MPC, VC, MEB. Data curation: RM, SZ, VC, MEB. Formal analysis: RM, VC, MEB. Funding acquisition: LS, AM, VC, MEB. Investigation: RM, FDM, CP, ML, FB, FCo, PC, RE, MS, FCa, VB, FS, LS. Methodology: RM. Project Administration: MEB. Resources: MC, AP, RB, AC, OR. Software: SZ. Supervision: AM, VC, MEB. Writing: MR, AR, AM, MPC, VC, MEB.

## Conflict of interest

MEB is founder and part owner of HMGBiotech, and MC and AP were partially supported by HMGBiotech. The other authors declare no conflict of interest.

## For more information

www.fondazionebuzziunicem.org
www.airc.it

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
