## [Review Process File · EMBO Molecular Medicine]

CXCR4 engagement triggers CD47 internalization and antitumor immunization in a mouse model of mesothelioma

Rosanna Mezzapelle, Francesco De Marchis, Chiara Passera, Manuela Leo, Francesca Brambilla, Federica Colombo, Maura Casalgrandi, Alessandro Preti, Samuel Zambrano, Patrizia Castellani, Riccardo Ertassi, Marco Silingardi, Francesca Caprioglio, Veronica Basso, Renzo Boldorini, Angelo Carretta, Francesca Sanvito, Ottavio Rena, Anna Rubartelli, Lina Sabatino, Anna Mondino, Massimo P. Crippa, Vittorio Colantuoni, and Marco E. Bianchi

DOI: [10.15252/emmm.202012344](https://doi.org/10.15252/emmm.202012344)

Corresponding authors: Marco Emilio Bianchi (bianchi.marco@hsr.it) , Vittorio Colantuoni (colantuoni@unisannio.it)

Review Timeline:

Submission Date:	14th Mar 20
Editorial Decision:	8th Apr 20
Revision Received:	2nd Mar 21
Editorial Decision:	19th Mar 21
Revision Received:	25th Mar 21
Accepted:	26th Mar 21

Editor: Zeljko Durdevic

Thank you for the submission of your manuscript to EMBO Molecular Medicine. We have now heard back from the two referees who agreed to evaluate your manuscript. As you will see from the reports below, while referee #1 is overall supporting publication of your work, referee #2 highlights the interest of the study but also raises a number of concerns that should be addressed in a revision of the current manuscript. Particular attention should be given to addressing the BoxA therapeutic effects in a second in vivo model. Addressing the reviewers' concerns in full will further be necessary for consideration of your manuscript in our journal.

Acceptance of the manuscript will entail a second round of review. Please note that EMBO Molecular Medicine encourages a single round of revision only and therefore, acceptance or rejection of the manuscript will depend on the completeness of your responses included in the next, final version of the manuscript. For this reason, and to save you from any frustrations in the end, I would strongly advise against returning an incomplete revision.

We would welcome the submission of a revised version within three to six months for further consideration. However, we realize that the current situation is exceptional on the account of the COVID-19/SARS-CoV-2 pandemic. Please let us know if you require longer to complete the revision.

I look forward to receiving your revised manuscript.

***** Reviewer's comments *****

Referee #1 (Remarks for Author):

The study by Mezzapelle & D'Agostino et al. describes in a series of elegant in vitro experiments a novel mechanism, showing how the N-terminal fragment of the alarmin HMGB1, Box1, interacts with CXCR4 which in turn promotes internalization of the do not eat me signal CD47 and thereby enhances uptake of mouse mesothelioma cells by macrophages. They go on to show in the mouse mesothelioma model which they phenotypically and genetically characterized previously that, when treated with Box1 three days after transplantation into the peritoneum of immune-competent mice, tumor growth is affected and T cell priming against the xenogeneic surrogate tumor-antigen luciferase is enhanced.

I think this study describes an important novel mechanism involving several new interactions between Box1, CXCR4 and CD47, which they show in two independent models. This is an important finding that should be published.

Major concern

In my view the major limitation of the study in its present form is the interpretation of the in vivo model. In their previous work they have carefully shown that the MM model resembles the human disease in terms of morphology, karyotype, histopathology. However, the immune response to such a transplanted tumor, carrying for example bovine serum proteins when transplanted, in my view clearly does not "recapitulate the human disease". It is known that tumor transplantation induces artifacts.

For example a fraction of transplanted cells typically dies (Peters L.J., Hewitt HB. The influence of fibrin formation on the transplantability of murine tumour cells: implications for the mechanism of the Revesz effect. Br J Cancer. 1974; 29:279-291). Thus, the mouse may be immunized before the tumor properly starts growing.

Therefore, claiming on page 8, second paragraph that "MM cells evoke a spontaneous immune response" is overstating the findings because transplantation artifacts can not be excluded.

My major concerns can be addressed by more critically presenting and discussing the data and do not call for more experiments. Also, I think it is sufficient to show the in vivo results in only one well described autochthonous model system.

Minor points:

1. Terminology: page 7, 3rd paragraph, they describe luciferin as a tumor associated antigen.

Tumor associated antigens, however, are by definition "antigens that are encoded typically by non-mutant cellular genes and expressed not only by cancer but also by at least some normal adult cells. Therefore, these antigens are not tumor specific and are commonly referred to as tumor associated antigens". Luciferase therefore is a tumor specific, xenogeneic, surrogate tumor antigen.

2. M+M section "In vivo CD8 depletion": please indicate the antibody clone used for depletion (2.43?) and the clone used for FACS analysis to confirm depletion of CD8 positive T cells.

Referee #2 (Remarks for Author):

This paper describes, in the context of mesothelioma, a concept of 'Immunogenic Surrender', whereby CXCR4 engagement leads to CD47 internalisation and activation of an anti-tumour immune response. BoxA, a fragment of HMGB1, binds to CXCR4, triggering CD47 internalisation and tumour cell phagocytosis by macrophages. The data is in general convincing and well presented, although there are some key questions to address as detailed below:

In Fig S2, are there many fibroblasts seen in this model, as is the case in the human disease?

Rather than reference to a previous paper (ref 9), it would be better to see a direct comparison here of the efficacy of BoxA in immunocompetent vs deficient mice, using this model.

As well as testing depletion of CD8 cells (Fig 2), it would also be worth testing of depletion of CD4 or NK cells, to see if these immune subsets also play a therapeutic role. In particular, innate immunity may also be important, and the depletion of NK cells would address this. Macrophage depletion, though challenging, should also be considered.

Is it not surprising that BoxA inhibits MM cell proliferation in vitro, but does not slow down growth when pre-treated cells are inoculate in vivo (Fig 3) - what might the explanation for this be?

The shifts in the FACS plots in Fig 3A are not impressive. Was this a single experiment or representative, and what did the MFI data show? Where is the control of secondary antibody only?

Fig 4F is unconvincing. A better experiment would be to take mice, treat with BoxA (or not), and characterise the immune cell infiltrate at a specific time point, including macrophages. This would be a more robust test of whether BoxA increases intratumoural macrophage infiltration. Using appropriate FACS panels on the immune cells within disaggregated tumours, it would also be possible to look at other cell types of interest, such as dendritic cells.

Fig 5 is interesting in that it tests a second cell line, namely CT26. However, to conclude that BoxA depletes surface CD47 in 'multiple tumour cells' is not justified. As well as more mouse cell lines, did the authors test human cells?

Is BoxA therapeutic in a second mouse cancer model - in particular an i.p. model of CD26 would be worth testing?

More minor points are:

In the 'Paper Explained' section, the authors state that anti-CD47 antibodies have 'already proven their value in numerous clinical trials'. This is overstating the benefit of these drugs clinically and should be toned down.

In the introduction the authors state that MM is 'most representative' of chronic inflammation causing cancer. It is not necessarily the 'most', as some would argue eg chronic inflammatory bowel disease causing cancer is better characterised/more important.

There is an updated review of ICD which the authors may want to cite in place of ref 11.

Galluzzi et al. *J Immunother Cancer*. 2020 Mar;8(1). pii: e000337. doi: 10.1136/jitc-2019-000337. Review.

Can the authors expand on any perceived route to the application of BoxA as a therapeutic?

***** Reviewer's comments *****

We thank both Reviewers for their excellent work, and our detailed answer to their queries is listed in the point-point reply to each individual Reviewer.

In addition, we point out that we have replaced several experiments shown in the initial submission, because the inadequate record-keeping. Other authors have independently replicated each of the main experiments, which is further proof of the solidity of the results. We have also substituted some of the experiments intended to demonstrate that CXCR4 (and not other receptors) mediate CD47 internalization. Fig 4 shows that CD47 is internalized, and Fig 6 shows that CD47 is co-internalized with CXCR4, but not with RAGE or TLR4. Overall, we think the changes we introduced make the revised manuscript stronger and more streamlined.

Referee #1 (Remarks for Author):

The study by Mezzapelle & D'Agostino et al. describes in a series of elegant in vitro experiments a novel mechanism, showing how the N-terminal fragment of the alarmin HMGB1, Box1, interacts with CXCR4 which in turn promotes internalization of the do not eat me signal CD47 and thereby enhances uptake of mouse mesothelioma cells by macrophages. They go on to show in the mouse mesothelioma model which they phenotypically and genetically characterized previously that, when treated with Box1 three days after transplantation into the peritoneum of immune-competent mice, tumor growth is affected and T cell priming against the xenogeneic surrogate tumor-antigen luciferase is enhanced.

I think this study describes an important novel mechanism involving several new interactions between Box1, CXCR4 and CD47, which they show in two independent models. This is an important finding that should be published.

We thank the Reviewer for his/her positive comments, which correctly capture the meaning and impact of our work.

Major concern

In my view the major limitation of the study in its present form is the interpretation of the in vivo model. In their previous work they have carefully shown that the MM model resembles the human disease in terms of morphology, karyotype, histopathology. However, the immune response to such a transplanted tumor, carrying for example bovine serum proteins when transplanted, in my view clearly does not "recapitulate the human disease". It is known that tumor transplantation induces artifacts.

We agree with the Reviewer and recognize that transplantable tumor models do not fully recapitulate autochthonous tumor development. To acknowledge the use of the transplantable cell line, we have revised the manuscript and now refer to the model as to "an immune-competent mouse model of human MM".

For example a fraction of transplanted cells typically dies (Peters L.J., Hewitt HB. The influence of fibrin formation on the transplantability of murine tumour cells: implications for the mechanism of the Revesz effect. Br J Cancer. 1974; 29:279-291). Thus, the mouse may be immunized before the tumor properly starts growing.

Therefore, claiming on page 8, second paragraph that "MM cells evoke a spontaneous immune response" is overstating the findings because transplantation artifacts cannot be excluded.

The Reviewer has a good point. In fact, what we meant was “Overall our results show that transplantation of MM cells evokes a spontaneous protective immune response”, where “transplantation of” was omitted in the original submission. We have now corrected the manuscript.

Also, while the advantage of their tumor model is that it grows autochthonous (arose in the peritoneum, transplanted into the peritoneum), starting the treatment on day 3 seems very early. It is not clear to me if Box1 interferes with establishing the proper environment for tumor growth or whether they actually treat an "established tumor" and induce a protective immune response to full blown MM, as the authors claim in the first sentence of the discussion? (for more information see also Wen F.T. A systematic analysis of experimental immunotherapies on tumors differing in size and duration of growth. *Oncoimmunology*. 2012 Mar 1;1(2):172-178.). Or in other words, what happens if you start treating on day 10?

The Reviewer has pinpointed an important limitation of our study, which we now explicitly acknowledge at page 13 in the Discussion. However, we do have addressed experimentally the question of when the MM tumor starts growing, or at least emits more photons. We followed the transplanted cells by BLI every few hours, see the attached figure. The BLI signal decreases slightly in the first few hours, but clearly increases in days 2 and 3. Thus, BoxA is administered when the tumor is actively growing. We show these data for the Reviewer only, but we could add them to our manuscript if you think it would be useful.

My major concerns can be addressed by more critically presenting and discussing the data and do not call for more experiments. Also, I think it is sufficient to show the in vivo results in only one well described autochthonous model system.

Minor points:

1. Terminology: page 7, 3rd paragraph, they describe luciferin as a tumor associated antigen. Tumor associated antigens, however, are by definition "antigens that are encoded typically by non-mutant cellular genes and expressed not only by cancer but also by at least some normal adult cells. Therefore, these antigens are not tumor specific and are commonly referred to as tumor associated antigens". Luciferase therefore is a tumor specific, xenogeneic, surrogate tumor antigen.

We thank the Reviewer for pointing out this important difference. We now use the terminology “surrogate tumor-associated antigen”.

2. M+M section "In vivo CD8 depletion": please indicate the antibody clone used for depletion (2.43?) and the clone used for FACS analysis to confirm depletion of CD8 positive T cells.

The antibody for depletion was the CD8 monoclonal antibody Clone 2.43, Bio X Cell. For CD8 staining, we used antibody clone 53-6.7, Biolegend. We have corrected the text.

Referee #2 (Remarks for Author):

This paper describes, in the context of mesothelioma, a concept of 'Immunogenic Surrender', whereby CXCR4 engagement leads to CD47 internalization and activation of an anti-tumor immune response. BoxA, a fragment of HMGB1, binds to CXCR4, triggering CD47 internalization and tumor cell phagocytosis by macrophages. The data is in general convincing and well presented,

We thank the Reviewer for his/her appreciation.

although there are some key questions to address as detailed below:

In Fig S2, are there many fibroblasts seen in this model, as is the case in the human disease?

In a previous paper, we had checked for the presence of fibroblasts in AB-derived tumors by staining for alpha-smooth muscle actin (α -SMA) (Fig 9, Mezzapelle et al. Sci Rep 2016). All three histological subtypes of mesothelioma derived from AB cells contain numerous cells that express α -SMA (brown). Data are depicted here for Reviewer's use and referenced to within the revised manuscript.

Rather than reference to a previous paper (ref 9), it would be better to see a direct comparison here of the efficacy of BoxA in immunocompetent vs deficient mice, using this model.

We agree with the Reviewer that a direct comparison would have been better, but we respectfully note that the depletion of CD8 cells (Fig 2) in its own way constitutes an immunodepleted mouse model. For this reason, we decided to focus on the setting up of a new tumor model, and on the role of macrophages.

As well as testing depletion of CD8 cells (Fig 2), it would also be worth testing of depletion of CD4 or NK cells, to see if these immune subsets also play a therapeutic role. In particular, innate immunity may also be

important, and the depletion of NK cells would address this. Macrophage depletion, though challenging, should also be considered.

We agree that testing the depletion of CD4 and NK cells would have been informative, and we plan to perform these experiments in the near future. We decided to prioritize the analysis of macrophages, for their central role in the proposed mechanism of Immunogenic Surrender. We thus tested the impact of their depletion with an anti-CSF1R antibody. The results are shown below.

Figure for referees not shown.

We were successful in depleting a large proportion of macrophages from the tumor (panels B-E). BLI (panel F) and survival curves (panel G) indicate that macrophages in fact support MM growth, and that their depletion extends the survival of mice. This is in agreement with many findings in disparate tumor models that Tumor Associated Macrophages support tumor growth. Treating macrophage-depleted mice with BoxA does not afford any advantage, both in terms of tumor growth or survival. Thus, removing macrophages makes BoxA ineffective, which is in accordance with our hypothesis that macrophages are needed for the mode of action of BoxA. However, we realized that since macrophage depletion *already* confers some advantage in MM mice, our experiment is potentially underpowered to prove an *additional* advantage (or lack thereof) of BoxA on top of that of the anti-macrophage antibody alone. Moreover, a small additional advantage of BoxA treatment, if indeed it exists, might be due to the direct effect of BoxA on tumor cells, as we had previously proven in the xenograft model (Yang et al, 2015) and in the present work (Figure 3E and 5E). Given that our experiment is strongly suggestive, but not definitive, we would prefer not to include it in the present revision, but to use these results in future work where we would also like to address whether macrophages are solely responsible for Immunogenic Surrender, or dendritic cells also play a role.

Is it not surprising that BoxA inhibits MM cell proliferation in vitro, but does not slow down growth when pre-treated cells are inoculate in vivo (Fig 3) - what might the explanation for this be?

The experiment reported in Fig. 3 was done to prove that the cells treated with BoxA do not behave as cells undergoing ICD. MM cells were treated with 800 nM BoxA only once, and then washed before inoculation. They thus grew as a tumor just as untreated cells. Indeed, only cells treated with an ICD inducer would grow less or not at all, because a large fraction would have undergone apoptosis. We thus think that the lack of response of MM cells to a single short-term exposure to BoxA before inoculation is fully justified.

The shifts in the FACS plots in Fig 3A are not impressive. Was this a single experiment or representative, and what did the MFI data show? Where is the control of secondary antibody only?

In fact, in the original submission we had omitted the control of secondary antibody only. In the present revision, we redid this experiment using a fluorescent primary antibody, and an isotype fluorescent antibody was used as control; the legends to Fig 3A and EV2A have been modified accordingly, as well as Material and Methods. We agree that the shifts are not impressive, but we repeated the experiment many times with different cell lines, and the result is always consistent.

Fig 4F is unconvincing. A better experiment would be to take mice, treat with BoxA (or not), and characterise the immune cell infiltrate at a specific time point, including macrophages. This would be a more robust test of whether BoxA increases intratumoural macrophage infiltration. Using appropriate FACS panels on the immune cells within disaggregated tumours, it would also be possible to look at other cell types of interest, such as dendritic cells.

We thank the Reviewer for the comment and indeed we attempted the suggested analysis. Unfortunately, to date, results remain inconclusive because of the high variability from mouse to mouse of the tumor infiltrating cells. We accept that the experiment in Fig 4F needs further validation; for this reason, we have preferred to remove it from the current revised manuscript.

Fig 5 is interesting in that it tests a second cell line, namely CT26. However, to conclude that BoxA depletes surface CD47 in 'multiple tumour cells' is not justified. As well as more mouse cell lines, did the authors test human cells?

We followed the Reviewer's suggestion and have now tested multiple mouse and human cell lines (mouse colon carcinoma MC38, mouse melanoma B16 and human glioblastoma U87) which respond to BoxA with CD47 surface depletion (Figure EV4A) and in the human colon adenocarcinoma cell line LoVo (Figure EV4B-E).

Is BoxA therapeutic in a second mouse cancer model - in particular an i.p. model of CD26 would be worth testing?

This was a very interesting suggestion, and we focused on setting up a second model using the CT26 cell line. We transplanted CT26 intraperitoneally rather than subcutaneously, and CT26 invaded the intestine. This is more in line with their original location, since CT26 cells derive from a colon carcinoma. The results fully confirm that BoxA has a therapeutic effect in this model as well (new Fig 5).

More minor points are:

In the 'Paper Explained' section, the authors state that anti-CD47 antibodies have 'already proven their value in numerous clinical trials'. This is overstating the benefit of these drugs clinically and should be toned down.

We have toned down our statement to "anti-CD47 antibodies ... are in advanced clinical development".

In the introduction the authors state that MM is 'most representative' of chronic inflammation causing cancer. It is not necessarily the 'most', as some would argue eg chronic inflammatory bowel disease causing cancer is better characterised/more important.

The Reviewer is obviously right. We have now stated that mesothelioma is "as representative" as colon carcinoma.

There is an updated review of ICD which the authors may want to cite in place of ref 11.

Galluzzi et al. J Immunother Cancer. 2020 Mar;8(1). pii: e000337. doi: 10.1136/jitc-2019-000337. Review.

Thank you for the suggestion; we were not aware of this paper at the time of submission. We have now cited this publication.

Can the authors expand on any perceived route to the application of BoxA as a therapeutic?

We have better discussed the potential development of BoxA as a therapeutic in the Discussion:
“In the perspective of drug development, BoxA can be considered a hit molecule that enabled the discovery of Immunogenic Surrender; such a hit might be improved by design or entirely replaced by small molecules that bind to CXCR4 and promote CD47 internalization, while not retaining the full agonist activity of CXCL12.”

Thank you for the submission of your revised manuscript to EMBO Molecular Medicine. Also, thank you for performing new experiments to confirm the results for which the record-keeping was compromised. I am pleased to inform you that we will be able to accept your manuscript pending the following final amendments:

1) In the main manuscript file, please do the following:

- Correct/answer the track changes suggested by our data editors by working from the attached/uploaded document.
- Add up to 5 keywords.
- Make sure that all special characters display well.
- In M&M, the statistical paragraph should reflect all information that you have filled in the Authors Checklist, especially regarding randomization, blinding, replication.
- Add contributions for Chiara Passera and Manuela Leo. Please use authors' initials instead of full names and the CRediT contributor role taxonomy. Please check "Author Guidelines" for more information.

<https://www.embopress.org/page/journal/17574684/authorguide#authorshipguidelines>

- Add data availability statement. If no data are deposited in public repositories, please add the sentence: "This study includes no data deposited in external repositories". Please check "Author Guidelines" for more information.

<https://www.embopress.org/page/journal/17574684/authorguide#availabilityofpublishedmaterial>

2) Appendix: Please provide a .pdf file with table of content and figure legend.

3) Movies: Please rename movie to "Movie EV1", remove its legend from the manuscript main file and zipp it as a .doc file with the movie file.

4) Synopsis image: Please provide image as a high-resolution .jpeg file 550 px-wide x (250-400)-px high.

5) Synopsis text: Every published paper now includes a 'Synopsis' to further enhance discoverability. Synopses are displayed on the journal webpage and are freely accessible to all readers. They include a short stand first (maximum of 300 characters, including space) as well as 2-5 one sentence bullet points that summarise the paper. Please write the bullet points to summarise the key NEW findings. They should be designed to be complementary to the abstract - i.e. not repeat the same text. We encourage inclusion of key acronyms and quantitative information (maximum of 30 words / bullet point). Please use the passive voice.

6) For more information: There is space at the end of each article to list relevant web links for further consultation by our readers. Could you identify some relevant ones and provide such information as well? Some examples are patient associations, relevant databases, OMIM/proteins/genes links, author's websites, etc...I

***** Reviewer's comments *****

Referee #1 (Comments on Novelty/Model System for Author):

The study and manuscript by Mezzapelle has been extensively revised all questions raised have been adequately addressed.

Referee #1 (Remarks for Author):

The study and manuscript by Mezzapelle has been extensively revised. It should be emphasized that in the new study, the effect of BoxA was analyzed in a second model. The chemically (N-nitroso-N-methylurethane) induced colon tumor 26 was now tested in vivo (that cell line was previously only analyzed in vitro). Furthermore, the in vitro analyses were extended to other cell lines.

In vivo, in contrast to the mesothelioma model in which 19 of 20 animals show tumor regression after treatment, the effects of BoxA treatment of CT26 are less pronounced. Only two of the 11 CT26 treated animals show a slight tumor growth retardation and one of these 11 animals rejects the tumor. I think the new data give a more holistic picture of (in this case the heterogeneous effect) of the new treatment approach. The in vitro studies further elucidate the mechanistic relationship. Overall, I stand by the earlier assessment that the authors have demonstrated an important new mechanistic link between BoxA, CXCR4 and internalization of the don't eat me signal CD47. In my view, the significance of the observation is so important that the paper should be published in EMMM in its newly revised form. The authors have adequately addressed all the important points raised during the first revision. In my first review, I had urged for a somewhat more critical discussion and a more restrained interpretation; the authors have moderately accommodated this request and, I think the manuscript can be published in its present form.

Since EMMM publishes the reviewers' comments, I would like to take this opportunity to once again clearly state my position as formulated in the first review: I still consider it misleading to write about immune surveillance and anti-tumor T cell priming (last sentence of the Introduction) because for such statements antigen-specific T cell responses need to be investigated. The first paragraph of the discussion suggests that BoxA initiates a productive T cell-mediated immune response through the newly postulated process of "immunogenic surrender." I think it is more likely that the tumor transplant artifact and the dying tumor cells have successfully primed T cells, which are then as effector cells better activated or sustained by the new BoxA treatment (immunogenic surrender). This in the future could be easily studied in the case of CT26, looking at responses towards the immune dominant rejection antigen AH1 (an H2-Ld-restricted peptide AH1 (SPSYVYHQF) which is derived from an activated endogenous retrovirus in CT26 cells, see also Huang, A.Y., et al. 1996. Proc. Natl. Acad. Sci. USA. 93:9730-9735). The investigation of antigen-specific T cell responses against AH1 (SPSYVYHQF) could clarify in follow-up studies of the authors whether T cells induced by "tumor transplantation immunization" are supported in their activity by BoxA, or whether, as the authors assume, BoxA induces the "immune surveillance". It remains to be seen whether BoxA is able to overcome the profound T cell tolerance of most autochthonous tumors. These comments are clearly not meant to ask for more experiments after this long and exhaustive review process, but rather as a critical comment concerning the interpretation and discussion of the results. I am curious to see further work on the authors' newly discovered mechanism.

Referee #2 (Comments on Novelty/Model System for Author):

The testing of a second model in this revised manuscript is important and reassuring.

Referee #2 (Remarks for Author):

The authors have performed a significant amount of work which, for the most part, addresses my concerns.

The depletion of macrophages making treatment with BoxA ineffective is an important experiment. However, the authors do not wish to add this data to the current manuscript (they would rather use it within future studies). They argue that this is appropriate because macrophage depletion without any treatment increases tumour growth, so they cannot pick apart the role of macrophage depletion on intrinsic tumour cell growth, relative to its impact on the efficacy of treatment. Overall, I think this is reasonable and am persuaded by their argument.

The testing of more mouse and human cell lines is important and reassuring. Moreover, the confirmation of in vivo therapy in a second model, CT26, is significant and convincing. This new Fig 5 is the key additional data required, which leads me to recommend accepting this revised manuscript.

The authors performed the requested editorial changes.

We are pleased to inform you that your manuscript is accepted for publication and is now being sent to our publisher to be included in the next available issue of EMBO Molecular Medicine.

Corresponding Author Name: Marco Emiio Bianchi
Journal Submitted to: EMBO MOLECULAR MEDICINE
Manuscript Number: EMM-2020-12344-V2